# The recurring impact of storm disturbance on black sea bass (*Centropristis striata*) movement behaviors in the Mid-Atlantic Bight

Caroline J. Wiernicki[1]☉*, Michael H. P. O'Brien[1]‡, Fan Zhang[2,3]‡, Vyacheslav Lyubchich[1]‡, Ming Li[2]‡, David H. Secor[1]☉

**1** Chesapeake Biological Laboratory, University of Maryland Center for Environmental Science, Solomons, Maryland, United States of America, **2** Horn Point Laboratory, University of Maryland Center for Environmental Science, Cambridge, Maryland, United States of America, **3** Southern Marine Science and Engineering Guangdong Laboratory (Zhuhai), Zhuhai, China

☉ These authors contributed equally to this work.
‡ These authors also contributed equally to this work.
* cjwiernicki@gmail.com

**Data Availability Statement:** All .csv and .mat files are available from the Dryad database (https://datadryad.org/stash/share/OztyXXL-

## Abstract

Storm events are a significant source of disturbance in the Middle Atlantic Bight, in the Northwest Atlantic, that cause rapid destratification of the water column during the late summer and early fall. Storm-driven mixing can be considered as a seasonal disturbance regime to demersal communities, characterized by the recurrence of large changes in bottom water temperatures. Black sea bass are a model ubiquitous demersal species in the Middle Atlantic Bight, as their predominantly sedentary behavior makes them ideal for tagging studies while also regularly exposing them to summer storm disturbances and the physiological stresses associated with thermal destratification. To better understand the responsiveness of black sea bass to storm impacts, we coupled biotelemetry with a high-resolution Finite Volume Community Ocean Model (FVCOM). During the summers of 2016–2018, 8–15 black sea bass were released each year with acoustic transponders at three reef sites, which were surrounded by data-logging receivers. Data were analyzed for activity levels and reef departures of black sea bass, and fluctuations in temperature, current velocity, and turbulent kinetic energy. Movement rates were depressed with each consecutive passing storm, and late-season storms were associated with permanent evacuations by a subset of tagged fish. Serial increases in bottom temperature associated with repeated storm events were identified as the primary depressor of local movement. Storm-driven increases in turbulent kinetic energy and current velocity had comparatively smaller, albeit significant, effects. Black sea bass represents both an important fishery resource and an indicator species for the impact of offshore wind development in the United States. Their availability to fisheries surveys and sensitivity to wind turbine impacts will be biased during periods of high storm activity, which is likely to increase with regional climate change.

NTbIWF5i3FmdNLPNoreIwnZnDRQYgTnHJX8).
Note this is a temporary, private URL and a
permanent, publicly accessible DOI will be
generated pending potential manuscript
acceptance.

**Funding:** Funding was provided to D. H. Secor by
the Maryland Department of Natural Resources
(MD DNR) and the Maryland Energy
Administration under grant 14-16-2151 MEA, grant
14-17-2661 MEA, and grant 14-18-2415. The
funding agency MD DNR advised on study design
prior to data collection.

**Competing interests:** The authors have declared
that no competing interests exist.

# Introduction

Storm disturbance is a key structuring force in coastal marine ecosystems, affecting population and community dynamics, as well as the habitats upon which they depend [1, 2]. For example, storms caused significant shifts in species composition and abundance for fish communities inhabiting shallow mangrove habitats [3], and increased frequency of high-intensity storms has been linked to decreased fish abundance and changes in trophic structure in kelp forests [4, 5]. However, impacts of storm disturbance on fish communities in deeper offshore marine ecosystems has not received as much attention, as impacts are more difficult to observe and presumably because such systems may be better buffered owing to their depth and volume.

A small but growing pool of research has emerged emphasizing the role of storms as singular, extreme disturbances driving changes in movement behaviors by marine fishes. Biotelemetry studies off the coasts of Florida and North Carolina observed storm-driven evacuation by tagged juvenile blacktip sharks, *Carcharhinus limbatus*, associated with decreased barometric pressure [6] and gray triggerfish, *Balistes capriscus*, associated with increased wave orbital velocity [7]. Storm-driven decreases in temperature, dissolved oxygen, and salinity were observed to drive emigration of striped bass, *Morone saxatilis*, from the Hudson River Estuary to coastal habitats [8]. Summer flounder, *Paralichthys dentatus* [9], and black sea bass, *Centropristis striata* [10], evacuations occurred following severe storm events in the United States mid-Atlantic. Hurricane disturbance on reef communities has also driven decreased movement and tighter coupling of fish to structured habitat [11–13]. Marine community responses to episoidic storm events represents challenging field science, perhaps contributing to a lack of studies on multiple storm events as a recurring source of natural disturbance. The vast majority of storm impacts on marine communities focus on the effects of storms as a singular, passing stochastic event. There is a critical gap in the literature regarding the potential of repeated storm events to serve as a natural, recurring disturbance regime, driving acute shifts in marine habitat conditions and incurring subsequent behavioral changes in habitat use and movement patterns from the affected fish communities.

The Mid-Atlantic Bight (MAB)—the continental shelf extending from the southern flank of Georges Bank offshore of Massachusetts, to Cape Hatteras, North Carolina—is a region regularly susceptible to significant storm disturbance during the summer and fall months. Storms can drive a number of changes in the physical environment on the shelf, such as changes in sea surface temperature due to vertical mixing [14–17]; increased turbulence at surface and bottom-boundary layers, either due to wind-driven shear or stirring [14, 16, 17]; and increased current velocity gradients in the oceanic surface and mixed layers [14, 15]. The MAB is also vulnerable to storm-driven temperature disturbance due to the presence of an oceanographic feature known as the "cold pool." The cold pool is an isolated layer of relatively colder, saltier bottom water within the MAB, receiving winter waters formed at the Nantucket Shoals [18, 19]. It forms seasonally with the vernal heating of surface waters (May-July), resulting in the stratification of the water column [18, 20]. This stratification and associated bottom-layer cold pool can be rapidly "destroyed" through the wind-driven fall overturn [21–23] commonly precipitated by late summer (mid- to late-August) and fall (September-October) storms [23–25].

Summer and fall storm events, such as tropical cyclones, contribute to the seasonal deterioration of the cold pool through wind-driven mixing and advection; storm-driven destratification events can create bottom temperature increases as high as 10°C over 24 hrs during a single hurricane event [10]. In this capacity, summer (June-August) and fall (September-October) storms in the MAB act as a significant source of natural disturbance by driving rapid partial-to-total destratification of the cold pool. Partial destratification occurs when mixing is

transitory and the water column restratifies, typically within days, whereas total destratifica-tion, a.k.a. cold pool destruction, is permanent [10].

Storm events in the MAB expose demersal fish communities to multiple physical and physi-ological stressors. For many fish species, chronic increases in temperature have been linked to behavioral impacts such as delayed reaction speed [26] and decreased movement related to energy and ventilation demands under thermal stress [27]. Acute increases in temperature have also been linked to increased discrete physiological impacts for a variety of species, such as rapidly increased cardiac output and decreased blood-oxygen binding in cod [28], reduced cardiac action potentials in carp [29], and upregulated gene expression of heat-shock and cell-cycle arrest proteins in gobies [30]. For black sea bass (300 g), increases in temperature from 12 to 24˚C, resulted in a 2-fold increase in standard metabolism and a 33% decline in aerobic scope [31]. Therefore, the rapid changes in bottom water temperature in the MAB are likely a significant source of disturbance and physiological stress to demersal fishes.

The rapid increase in bottom water temperatures owing to storm mixing co-occurs with rapid changes in current velocity, turbulent kinetic energy, and noise. Increased storm-related flow and turbulence have been demonstrated to elicit elevated movement responses in an oce-anic fish species [7] and various coastal sharks [32]. Storm-generated noise has been demon-strated to occur within low-range frequency bands [33, 34] that also overlap with the majority of fish hearing detection ranges [35]. Several storm systems within a year can cause repeated thermal destratification events, as well as simultaneously increasing current and turbulent flow within the water column. As such, the cumulative impacts caused by these storm-driven stress-ors—seasonal storm-driven fluctuations in temperature and flow—could represent a dynamic disturbance regime within the MAB, influencing its demersal fish communities.

Here, we investigate the effect of storm events as a recurring source of disturbance to a common member of the demersal MAB shelf assemblage: black sea bass. Black sea bass are a mostly sedentary, reef-associated species [36–39] that exhibit an affinity for both artificial and natural structure [40], making them an ideal candidate for biotelemetry studies on potential shifts in fish movement behaviors. Black sea bass and other demersal fish assemblages in the MAB characteristically occupy nearshore shelf habitats from late spring through fall; during the late fall, they then undertake cross-shelf migrations to deeper waters, typically throughout mid-September to late October [37, 38, 41, 42]. This transit to off-shelf water succeeds the arrival of late summer and early fall hurricanes and tropical storms in the western Atlantic. Secor et al. [10] hypothesized that cumulative storm impacts during this late summer and early fall period could cue offshore seasonal migration. The potential role of late summer-fall storm disturbances in the MAB to serve as a migratory cue emphasizes the need to understand repeated storms as an ecologically-significant disturbance regime.

Comprehensive understanding of this natural disturbance regime shaped by storms is both timely and relevant to the MAB, as the region is currently undergoing evaluation by industry and policy stakeholders for offshore wind energy development [43]. Black sea bass is a model species for understanding wind energy impacts because they are ubiquitous and commercially important within the $> 7 \cdot 10^3$ km$^2$ of leased USA federal waters (https://www.boem.gov/renewable-energy/state-activities). To best utilize this and similar demersal species in under-standing both negative and positive impacts of offshore wind energy development, baseline information is needed on storm impacts to black sea bass. Storm effects on fish behavior are a pervasive recurrent natural disturbance in the MAB, which if not understood and accounted for, will likely bias wind turbine impact studies.

The goal of this study was to better characterize the recurring impact of seasonal storms on the local oceanography in the MAB, and their subsequent recurring impact on movement and evacuations by black sea bass. We hypothesized that: (1) storm events are a recurring feature

that impact black sea bass habitat through changes in temperature, bottom current velocity, and turbulent kinetic energy; (2) changes in movement behavior are caused by both individual and cumulative storm-driven environmental changes; and (3) storm-related movement behaviors are driven chiefly by rapid (<1 d) mixing and increased bottom temperature. During three summer-fall seasons (2016–2018), we measured evacuation and movement behaviors through biotelemetry, coupling these behaviors with predicted storm-driven changes in water column conditions provided by a coastal ocean model. In particular, the year 2017 provided a unique opportunity to model cumulative storm impact, as multiple storm events occurred with sequential impacts on the water column during a single season.

## Materials and methods

This study was approved by University of Maryland Center for Environmental Science (UMCES) Institutional Animal Care and Use Committee (IACUC) (Protocol Number F-CBL-16-10), under UMCES IACUC Chair Dr. Christopher L. Rowe. All surgery was performed under Aqui-S anesthetic (20 mg/L, active ingredient clove oil) to minimize suffering and injury to fish.

### Study site

This project included three study reefs located 16–46 km east and southeast of Ocean City, Maryland, USA (Fig 1, Table 1), which were respectively identified as the Northern, Middle, and Southern sites. All study reefs overlapped with the presence of the cold pool for approximately 4–5 months each year (spring-late summer) and ranged in depth from 20 m to 27 m (Table 1).

**Acoustic telemetry data collection.** A total of nine VEMCO VR2AR (VEMCO Ltd.) acoustic-release receivers were deployed across study sites during June-October, 2016–2018 (Table 1). Three receivers at each site were positioned to capture movement behaviors. Receivers were deployed at an 800 m distance and at 0˚, 120˚, and 240˚ angles from capture and tagging locations at each reef. The 800 m distance was set based on detection ranges observed in a range test study under similar conditions off the coast of New Jersey [40]. Receivers were moored to the seabed with two 20.4 kg weight plates and positioned in the water column with one 10.8 kg-buoyant buoy each. Receivers continuously recorded data on unique transmitter detections, while recording bottom water temperature (˚C), and ambient noise at 69 kHz (mV) every 600 seconds.

Animal collection, surgical, anesthetic, and release procedures were approved by the UMCES Institutional Animal Care and Use Committee (IACUC-Secor-F-CBL-160-10). During June 2016–2018, 8–17 black sea bass at each site were surgically implanted with VEMCO V9-2H acoustic transmitters, which emitted a 69 kHz signal at randomized 90-second intervals (Table 1). Fish were captured at reef sites using rod-and-reel on a chartered recreational fishing boat and immediately placed in a 57-liter tank containing ambient seawater until surgery. Water within the tank was partially replaced at approximately 10-minute intervals to avoid deoxygenation. Sublegal (≤32 cm) individuals were selected for tagging in an effort to reduce transmitter loss from fishing mortality, as the reef sites are heavily fished by recreational anglers. Fish selected for surgery were transferred to a surgery tank containing a mixture of sea water and Aqui-S 20E anesthetic (AquaTactics; 20 mg $L^{-1}$; active ingredient clove oil). Once anesthetized, individuals were weighed using a spring-scale, measured for total length, and sexed. Sex was determined by the visual inspection and identification of gonads, which were visible during surgery through the incision (see below) [44]. Individuals for which sex could not be visually determined were recorded as of unidentified sex. Following the recording of

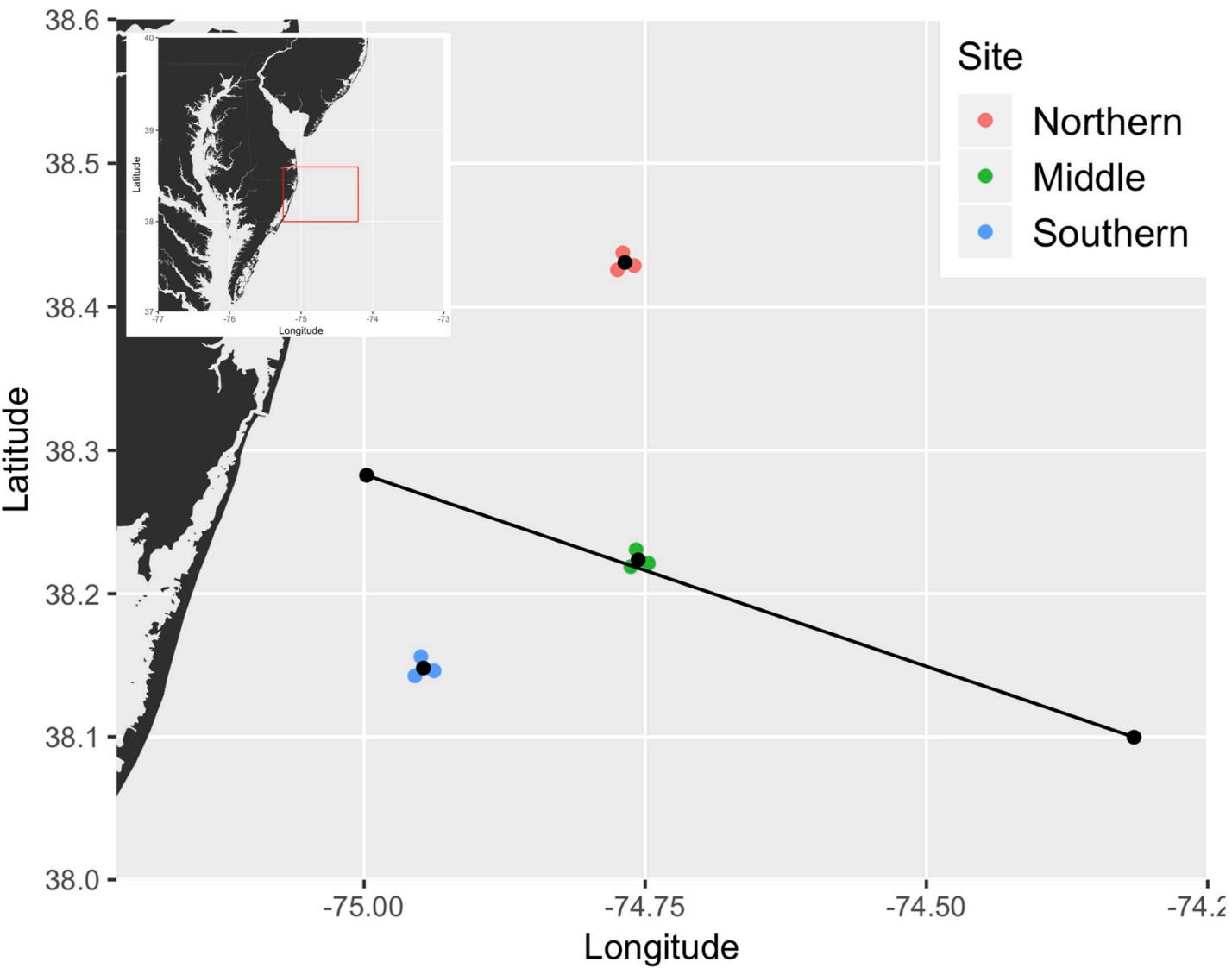

**Fig 1. Experimental reef sites, east of the Maryland coast, for the 2016–2018 study seasons.** * Colored points refer to receiver deployment locations, while black points refer to approximate tagging locations. The black line depicts the selected 38.73 km transect for cross-sectional FVCOM estimates. * Inset map provides location of study site, given as a red square, within the broader Maryland-Delaware coast.

body measurements, individuals were transferred to a sling lined with synthetic foam to minimize damage to fins and epithelium, and, while the head and gills remained immersed, a 1-cm incision was made lateral to the midline, preceding the vent. The transmitter was inserted through the incision and was closed with 1–2 single surgical-knot sutures. Post-surgery, fish were transferred back to the holding tank to monitor for recovery. At the Middle and Northern sites, barotrauma was observed owing to greater depths. Incisions alleviated internal pressure although barotrauma symptoms were likely not fully abated [42]. To promote recovery and reduce the risk of surface depredation by birds and large fishes, recovered fish were descended to half depth (~15 m) using a pressure-release device (Seaqualizer ©, http://seaqualizer.com/) at the site of their capture.

**Oceanographic model outputs.** A Finite Volume Community Ocean Model (FVCOM) was used to evaluate mesoscale oceanographic forcing on black sea bass movement. FVCOM

**Table 1. Summary of tagging location, quantity and condition characteristics of fish tagged, and receiver location and average depth.**

| Site | Year | Tagging Location | | Fish Tagged | | | Receiver Summary | | |
|---|---|---|---|---|---|---|---|---|---|
| | | Latitude | Longitude | N | Size (mm) | Weight (g) | Date Deployed | Date Retrieved | Average Depth (m) |
| Northern | 2016 | 38.4309 | -74.7680 | 15 | 260 ± 25 | 200 ± 90 | Jun 10 | Nov 2 | 26.07 |
| | 2017 | 38.4307 | -74.7677 | 15 | 262 ± 28 | 250 ± 70 | Jun 22 | Oct 26 | 25.27 |
| | 2018 | 38.4309 | -74.7680 | 27 | 251 ± 38 | 240 ± 70 | Jun 19 | Oct 23 | 25.82 |
| Middle | 2016 | 38.2237 | -74.7562 | 15 | 232 ± 30 | 260 ± 100 | Jun 12 | Nov 1 | 26.66 |
| | 2017 | 38.2234 | -74.7558 | 15 | 283 ± 17 | 320 ± 60 | Jun 22 | Oct 26 | 26.71 |
| | 2018 | 38.2237 | -74.7562 | 16 | 271 ± 37 | 280 ± 110 | Jul 15 | Oct 23 | 26.48 |
| Southern | 2016 | 38.1480 | -74.9472 | 15 | 267 ± 24 | 270 ± 80 | Jun 9 | Nov 1 | 20.95 |
| | 2017 | 38.1589 | -74.9439 | 8 | 256 ± 20 | 240 ± 50 | Jun 29 | Oct 26 | 22.54 |
| | 2018 | 38.1481 | -74.9472 | 17 | 241 ± 30 | 190 ± 60 | Jul 31 | Oct 26 | 21.05 |

*Only two receivers were recovered in 2017.

is a three-dimensional unstructured-grid hydrodynamic model that consists of momentum, continuity, temperature, salinity, and density equations [45, 46] and adopts Mellor-Yamada level 2.5 turbulent closure scheme. The model utilizes sigma-coordinate transformations and unstructured triangular cells to simulate flows along irregular coastlines, such as those prevalent in coastal shelf or estuarine systems like the MAB [45–47]. The model was configured for the MAB region, with the eastern boundary located approximately at 70° W, and the northern and southern boundaries located at approximately 42° N and 34° N, respectively. Initial conditions of salinity and temperature for the FVCOM were based on predictions from the Regional Ocean Modeling System (ROMS) Experimental System for Predicting Shelf and Slope Optics (ESPreSSO) model [48]. The FVCOM was run from January 1 to December 31 for each year. The open boundary conditions were prescribed using the temperature, salinity, and subtidal sea level from Hybrid Coordinate Ocean Model and Navy Coupled Ocean Data Assimilation systems (HYCOM-NCODA, http://hycom.org), and five major tidal constitutes ($M_2$, $S_2$, $K_1$, $O_1$, and $N_2$) from the Oregon State University TOPEX/Poseidon Global Inverse Solution TPXO 7.1 [49, 50]. The surface heat and momentum fluxes were prescribed using the North American Mesoscale (NAM) forecast system.

Time series data on modeled bottom water temperature, current velocity, and turbulent kinetic energy (TKE) were extracted at hourly intervals, for each receiver, for the duration of receiver deployment during each year of study for each site. Bottom water temperature was selected as an indicator of cold pool destruction/recovery and potential physiological stress; current velocity was selected as an indicator of physical forcing and potential physical stress owing to the need for increased energy devoted to maintaining position at reef habitat home ranges [7, 32]; and TKE was selected as an indicator of both destratification and vertical shear between water parcels, as well as an additional potential physical stress [7]. Time series of all three variables were averaged across receivers to yield averaged hourly predictions per site. Time series data on observed wind speed and direction were extracted from the North American Regional Reanalysis (NARR) products (https://psl.noaa.gov/data/gridded/data.narr.html) at 3-hr intervals.

A cross section of triangulated grid-point estimates of bottom water temperature, current velocity, and TKE was also obtained to investigate the response of the cold pool presence to identified storms. Estimated lateral outputs extended across the shelf in the Delaware-Maryland-Virginia peninsula region of the MAB; estimated cross-sectional measurements were taken along a 39 km bisection of the Middle study site (Fig 1). The model results were

compared with observed bottom water temperatures obtained at this study's acoustic receiver array (S1 Appendix) (S3A–S3C Fig).

**Storm identification.**   Observed peak winds, which are often used in storm warnings, varied substantially and did not convey information on storm duration. Therefore, storm presence and duration threshold were defined as the number of hours during which observed sustained wind speeds were > 5 m s$^{-1}$, although burst wind gusts identified during storm periods were considerably higher. The lower limit of 5 m s$^{-1}$ was selected to provide a conservative definition for potentially disruptive storm activity (3 or greater on the Beaufort Wind Scale [51]). Storms above this limit were further categorized and compared according to the Beaufort Wind Scale. Both named and unnamed events were considered. Following identification of storm presence and duration, modeled cross-shelf distributions of bottom water temperature, current velocity, and TKE were plotted and compared across the days before, during, and after each storm event.

**Data analysis: Movement behavior.**   Telemetry data were analyzed for changes in local and broad-scale movement behaviors relative to storm events. A movement index was estimated as the average number of movements detected by consecutive unique receivers per hr [40]. Hourly movement indices were aggregated across tagged fish within each site to provide a site activity index. Activity indices across sites were evaluated for each year, using an analysis of variance (ANOVA) test comparing activity indices across storm periods and nested by site. Each year an initial baseline period (no storm) was compared to subsequent storm periods; storm periods were defined as the period between onset of a particular storm and the onset of any ensuing storm. Post hoc multiple comparisons of activity indices across storm periods were conducted using Tukey contrasts. ANOVA tests and multiple comparisons were accomplished using the *car* [52], *lme4* [53], and *multcomp* [54] packages in R.

Broad-scale movements and subsequent departures from study sites (aka evacuations) were evaluated by calculating instantaneous and percent relative (the back-transformed rate of the instantaneous) tag loss rates, and by modeling the number of transmitters recorded per day using an autoregressive integrated moving average (ARIMA) intervention analysis [10]. Strong coastal storm events are capable of generating substantial noise owing to wind, wave action, or cavitation [33], which can diminish reception of transmitter signals and create "false" evacuations caused by acoustic interference. ARIMA intervention analysis facilitates the identification of false evacuations, identified as single points within the recorded time series that temporarily, but significantly, alter the behavior of the rest of the series. Significant outliers, data points that prevent the modeled time series from attaining stationarity, are identified through sequential t-tests. For this study, intervention analysis was applied to a time series of the number of unique transmitters recorded each day. The analysis tested for the presence of two types of interventions: (1) temporary shifts and (2) permanent level shifts. Permanent level shifts (stepped declines) are indicative of fish evacuation—interventions that fundamentally and permanently changed the remaining time series. Temporary shifts are those interventions that altered the time series temporarily and appear as nonlinear returns to the previous detection level (see Fig 9). When the ARIMA model did not converge on stationarity, departures owing to permanent or temporary shifts could not be accurately discriminated. This analysis was performed in R, using the *tsoutliers* package [55].

The relationship between local activity levels and individual storm variables (Table 2) was explored using a Generalized Additive Model for Location, Scale, and Shape (GAMLSS). Telemetry data and predicted FVCOM output supported analysis of the effects of multiple storm events and their cumulative impact as consecutive events for 2017 alone, as only a single storm event was identified in 2016 and 2018. The predictors were tested for their influence on the response variable, daily average movement index, and included daily average TKE,

**Table 2. Variables in black sea bass movement GAMLSS.**

| Variable | Type | Units | Source |
|---|---|---|---|
| Movement index | Numerical response | Average number of movements per unique receiver per hr | VEMCO VR2AR receiver array |
| Turbulent kinetic energy | Numerical predictor | Square meters per square second ($m^2\ s^{-2}$) | FVCOM prediction |
| Bottom water temperature | Numerical predictor | Degrees Celsius (˚C) | VEMCO VR2AR receiver array |
| Current velocity, differenced | Numerical predictor | Meters per second ($m\ s^{-1}$) | FVCOM prediction |
| Accumulated number of storm days within a calendar year | Numerical predictor | Total number of days | FVCOM prediction |
| Tagged individual, length | Numerical predictor | Millimeters (mm) | Tagged black sea bass measurement |
| Tagged individual, sex | Categorical predictor | Male (M), Female (F), Unidentified (U) | Tagged black sea bass measurement |
| Transmitter | Random effect | No units | VEMCO V9-2x acoustic transmitter ID |

observed daily average bottom water temperature, differenced modeled daily average current velocity, accumulated number of storm days (ANSD: the time series of total storm days throughout the study period), the sex of the tagged individual, and the length of the tagged individual. Unique transmitter code (individual fish) was incorporated as a random intercept, and lagged response variables were incorporated to account for temporal autocorrelation of the response. Site effects were tested separately in a nested ANOVA.

Prior to fitting the model, numerical variables were iteratively incorporated and compared in various model structures containing, raw, lagged, or differenced forms (i.e., TKE, bottom temperature, and current velocity) to assist with the detection and minimization of collinearity. Collinearity was identified based on calculation and comparison of variance inflation factors (VIF); variables with VIF > 10 were discarded. Although model parameters are necessarily related—particularly the FVCOM-derived current velocity and TKE variables—substituting and comparing lagged, differenced, and raw forms allowed greater differentiation of independence across these processes. All remaining, non-collinear numerical predictor variables were centered and scaled prior to incorporation in the model. Various models were constructed with different distributions and combinations of lagged response variables, and the final model was selected based on lowest Akaike information criterion (AIC).

In the GAMLSS, daily average movement index, $Y_{t,j}$, for day $t$ and fish $j$ was specified using a generalized gamma distribution, with the distribution mean, $\mu_{t,j}$, modeled as a linear combination of $k$ predictors using a logarithmic link function:

$$Y_{t,j} \sim \text{generalized gamma}(\mu_{t,j},\ \sigma_{t,j},\ v_{t,j})$$

$$\ln(\mu_{t,j}) = \beta_0 + \sum_{i-1}^{k} \beta_k X_{t,j,k} + \alpha_{t,j},$$

where $\beta_i$ ($i = 0, 1, \ldots, k$) are the fixed effects coefficients for the mean function, $\alpha_{t,j}$ is the random intercept for $j$th transmitter. No dependence on the predictors was specified for the scale parameter $\sigma$ and shape parameter $v$. Up to 400 iterations of the Rigby and Stasinopoulos algorithm were used to estimate the model parameters. All analysis for model development was completed in R, using the *car* [52], *lme4* [53], *gamlss* [56], and *forecast* [57, 58] packages.

## Results

### Storm events and destratification

Six storm events varying in timing (July-September), duration (33–87 hr), and intensity (maximum wind speed 13.4–16.6 m s$^{-1}$; Beaufort Wind Scale 6–7) occurred between the observed June-October study period of 2016–2018 (Table 3). The six storms were identified as (1) Tropical Storm Hermine, with peak wind speeds on September 3, 2016; (2) a nor'easter, with peak wind speeds on July 29, 2017; (3) Potential Tropical Cyclone 10 (PTC10), with peak wind speeds on August 30, 2017; (4) Tropical Storm Jose, with peak wind speeds on September 19, 2017; (5) Tropical Storm Maria, with peak wind speeds on September 27, 2017; and (6) an unnamed wind event, with peak wind speeds on September 9, 2018. Observed bottom water temperature showed rapid increases over the course of the first several hours following storm arrivals, indicative of wind-driven mixing, offshore advection, and destratification of the water column (Fig 2).

Patterns in observed bottom water temperatures showed a differential impact of storm-driven destratification across years and sites. In each 2016 and 2018, one significant storm disturbance was identified; TS Hermine in 2016 and an unnamed wind event in 2018 (Fig 2). During both years, cold pool temperatures remained relatively stable at 12.5–16.9°C with a gradual rise during August. Though moderate increases in temperature occurred prior to the large destratification events, the onset of storms caused permanent destratification and increases in temperatures that ranged from 5.7 to 10.9°C (8.9 [mean] ± 1.6°C [standard deviation]) between sites and years. During 2017, however, multiple storm events were identified: a destratification event at the end of July, recovery of stratification at two sites, then subsequent cycles of destratification and restratification through September. This pattern of decreasing bottom water temperature and restratification of the water column did not occur at the Southern site, which was also the shallowest (c. 21 m across years) and warmest site. Here, following August destratification, water temperatures remained elevated and the cold pool did not recover.

### Storm characterization: Modeled variables

Observed storms were categorized in terms of wind speed and duration of average wind speed, yielding two classes of comparatively stronger storms and more moderate storms. The stronger storms (longer duration, Beaufort Wind Scales mostly at 7) occurred in 2016 and 2017,

**Table 3. Mid-Atlantic Bight storm events, June-October 2016–2018.** All date-times are UTC.

| Year | Name | Arrival Date | Departure Date | Maximum Wind speed Date | Duration (hr) | Maximum Wind speed (m s$^{-1}$) | Mean Wind speed (m s$^{-1}$ ± SD) | Minimum Wind speed (m s$^{-1}$) | Beaufort Wind Scale |
|---|---|---|---|---|---|---|---|---|---|
| 2016 | Tropical Storm Hermine | Sep 2 09:00 | Sep 5 18:00 | Sep 3 21:00 | 81 | 15.25 | 8.28 ± 3.02 | 5.02 | 7 |
| 2017 | Nor'easter (unnamed) | Jul 29 15:00 | Jul 31 00:00 | Jul 29 18:00 | 33 | 14.79 | 10.25 ± 2.88 | 5.57 | 7 |
| | Potential Tropical Cyclone 10 | Aug 26 15:00 | Aug 30 12:00 | Aug 30 00:00 | 81 | 16.28 | 8.9 ± 2.75 | 5.1 | 7 |
| | Tropical Storm Jose | Sep 17 12:00 | Sep 20 15:00 | Sep 19 12:00 | 75 | 16.6 | 9.0 ± 3.36 | 5.23 | 7 |
| | Tropical Storm Maria | Sep 25 15:00 | Sep 29 06:00 | Sep 27 18:00 | 87 | 13.44 | 8.93 ± 1.96 | 5.23 | 6 |
| 2018 | Wind event (unnamed) | Sep 8 15:00 | Sep 10 12:00 | Sep 9 18:00 | 45 | 14.3 | 8.18 ± 3.06 | 5.05 | 7 |

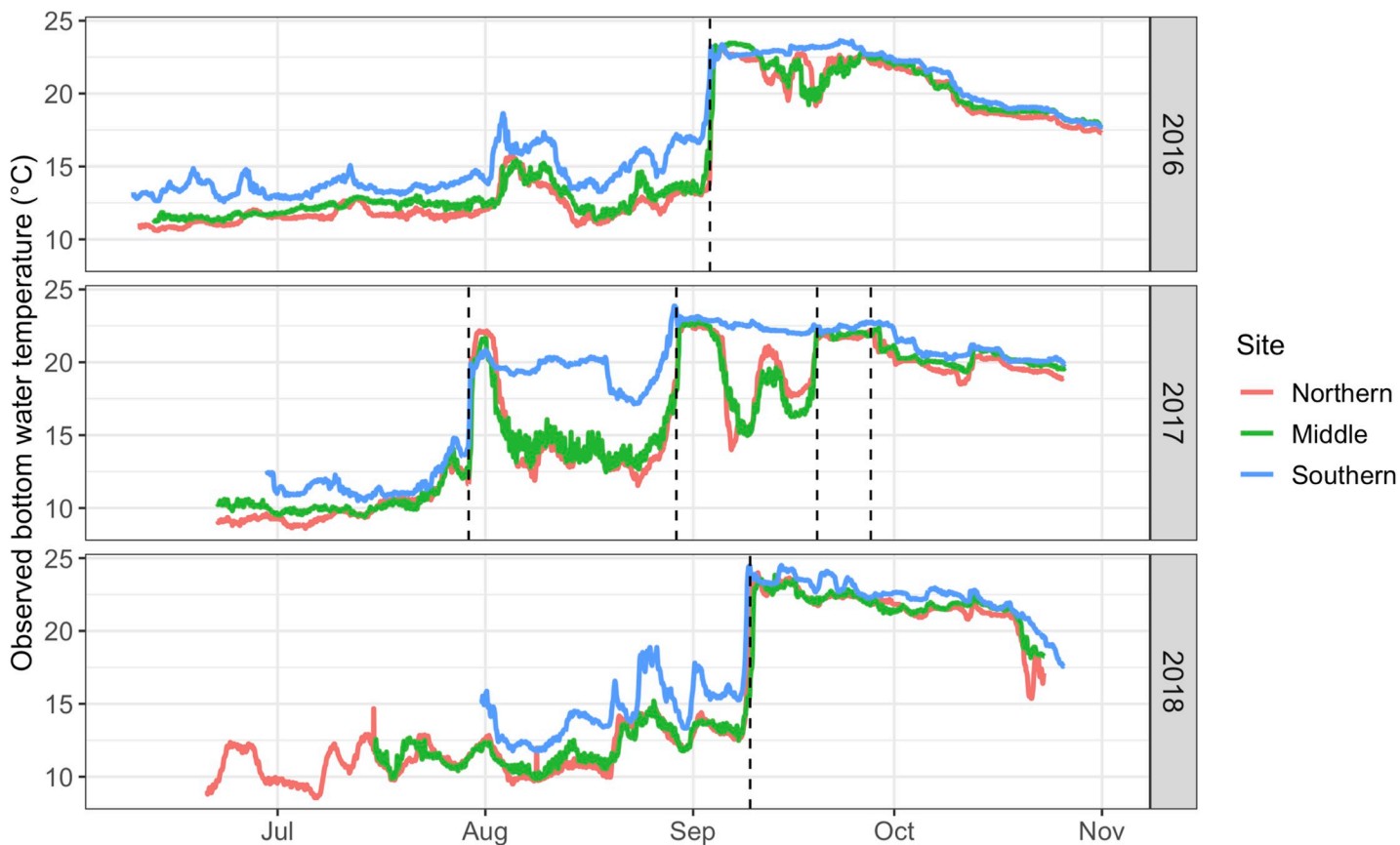

**Fig 2. Observed hourly bottom water temperature (˚C), averaged across each study site for each year.** * Black dashed lines refer to dates of observed maximum wind speed for identified storm events (see Table 3).

while 2018 experienced a comparatively moderate storm event, at nearly half the duration time and just barely clearing the wind speed necessary to reach Beaufort Wind Scale 7. In 2016 and 2017, TS Jose, PTC10, and TS Hermine brought in the highest wind speeds (16.6, 16.3, and 15.3 m s$^{-1}$, respectively; all Beaufort Wind Scale 7), while TS Maria, PTC10, and TS Hermine exhibited the longest duration (87, 81, and 81 hr, respectively) (Fig 3; Table 3). Conversely, the July nor'easter that occurred in 2017 reached a maximum wind speed of 14.8 m s$^{-1}$ (Beaufort Wind Scale 6) and lasted for only 33 hr; similarly, in 2018, the unnamed wind event reached peak wind speeds of 14.3 m s$^{-1}$ (low Beaufort Wind Scale 7) and continued 45 hr (Fig 3; Table 3). Across all years, however, modeled storm wind vectors indicated a predominance of northeasterly winds directed along shore.

The FVCOM predictions of hourly bottom water temperature, current velocity, and TKE peaked rapidly around periods of storm arrival and maximum wind speed (Fig 4; S1 and S2 Figs). The model successfully captured permanent destratification owing to storm events in 2016 and 2018, as well as the recovery and gradual increase in temperatures following repeated storm events in 2017 (Fig 4). Storm-driven increases in current velocity and TKE, on the other hand, were relatively high (velocity: 0.1–0.2 m s$^{-1}$; TKE: 0.0005–0.003 m$^2$ s$^{-2}$) and short-lived (10–25 hr), across all years, with little difference in baseline levels before and after storm events.

Storm-induced destratification events encompassed major portions of the shelf environment and spatial depictions of FVCOM outputs captured a range of destratification responses

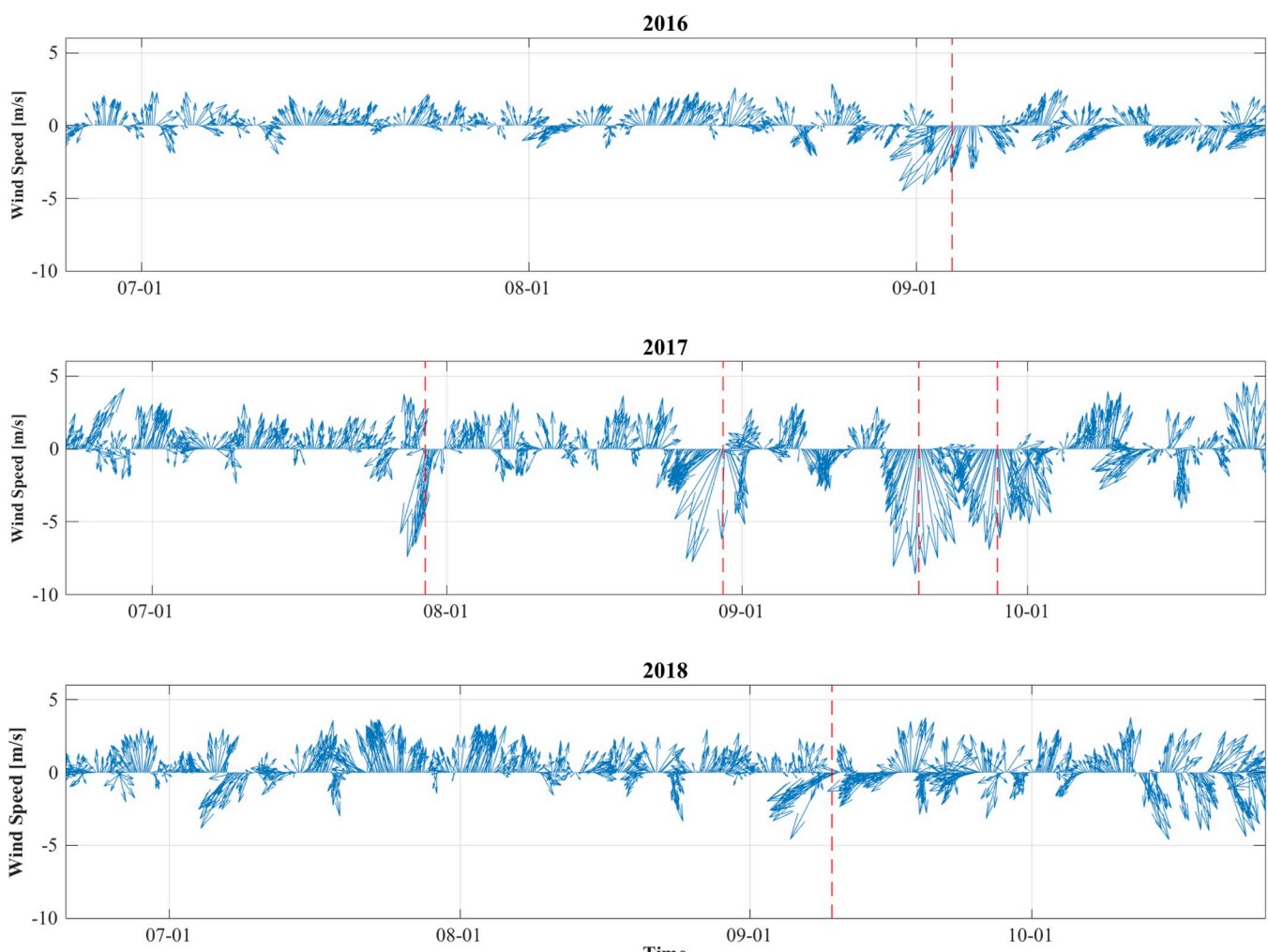

**Fig 3. Time series of hourly wind vectors (m s⁻¹) for 2016, 2017, and 2018.** * Red lines refer to the date and time (month-day hour) of peak wind speeds for each storm event (see Table 3).

to storms across the shelf's spatial extent and depth range (Figs 5–7). In years when single events caused permanent destratification (2016 and 2018), the nearshore water column became well-mixed after 1–2 days of the observed maximum wind speeds (Fig 5); the cold pool remained intact farther offshore, with inshore bottom waters increased by 10–15°C for days after the storm passed. Modeled bottom temperature for these single-storm years showed warmer temperatures extending towards mid-shelf waters (~30–35 m depth; Fig 6), corresponding to an offshore shift of the cold pool. The cold pool remained offshore for the rest of the late summer-fall season.

Summer 2017 exhibited a complex cycle of restratification (Fig 7). After the July nor'easter, stratification associated with the cold pool recovered to pre-storm levels; following PTC10, the cold pool recovered more slowly and to a lesser extent; the third storm—TS Jose—caused permanent destratification and destruction of the cold pool in waters < 60 km from shore. Similarly, bottom water temperatures for 2017 reflected a gradual retreat by the cold pool from shore (Fig 6), with permanent destratification occurring after TS Jose (Fig 7). Storm-driven current velocity and TKE decayed shortly after the passage of the storms (S4–S8 Figs).

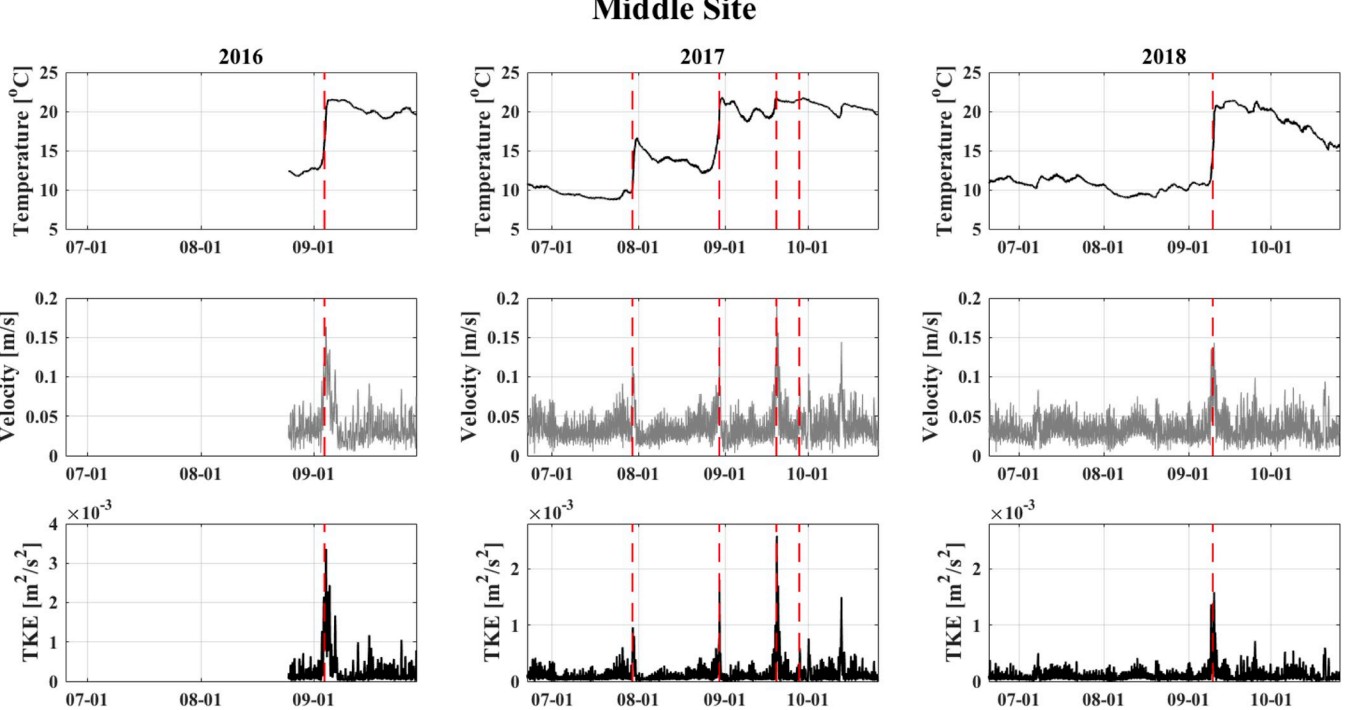

**Fig 4. The FVCOM estimates of hourly bottom water temperature (˚C), current velocity (m s$^{-1}$), and turbulent kinetic energy (TKE; m$^2$ s$^{-2}$), averaged for the Middle site for August-September 2016 and June-October 2017–2018.** * Dashed red lines refer to modeled maximum wind speeds occurring during each of the six identified storm events (see Table 3).

### Movement analysis: Observed activity and evacuation behavior

During all years, significant changes in local activity and evacuation rates from reef sites were observed in the wake of storms. Local activity indices were significantly different across each site before and after single storm events for each year (Fig 8). In 2016 and 2018, activity at all sites decreased by approximately 50%, following TS Hermine and the unnamed wind event, respectively, compared to the periods before these storms (ANOVA, p <0.001; Tukey, p < 0.001). Across all sites in 2017, activity indices were reduced by 50% between combined periods before and after PTC10 (ANOVA, p < 0.001; Tukey, p <0.001), with no other significant differences detected before and after the other storm events that year.

Transmitter loss over time indicated a steady decline in the number of unique tags present at each site over each year of study, which could be modeled through exponential decay (Table 4), ranging between <0.004% and 0.021% d$^{-1}$ across years and sites.

Significant increases in the number of fish evacuating reef sites were identified during days of peak wind speed for storm events during all years. Though we did observed instances of fish returning days to weeks following some storm events, evacuations from reef habitats were defined operationally as departures exceeding one week. ARIMA intervention analysis indicated permanent event-driven declines in fish presence at all sites during all years (Fig 9). Permanent level shifts were identified at the Northern and Middle sites during 2016, with the numbers changing considerably the day after TS Hermine's wind speeds peaked. In 2017, permanent level shifts were also identified during the July nor'easter (at the Middle and Southern sites), the PTC10 (at all sites), and immediately before peak winds arrived for TS Jose (at the Northern site). Lastly, a permanent level shift at the Southern site was identified during the date of maximum wind speed associated with 2018's unnamed wind event. The ARIMA

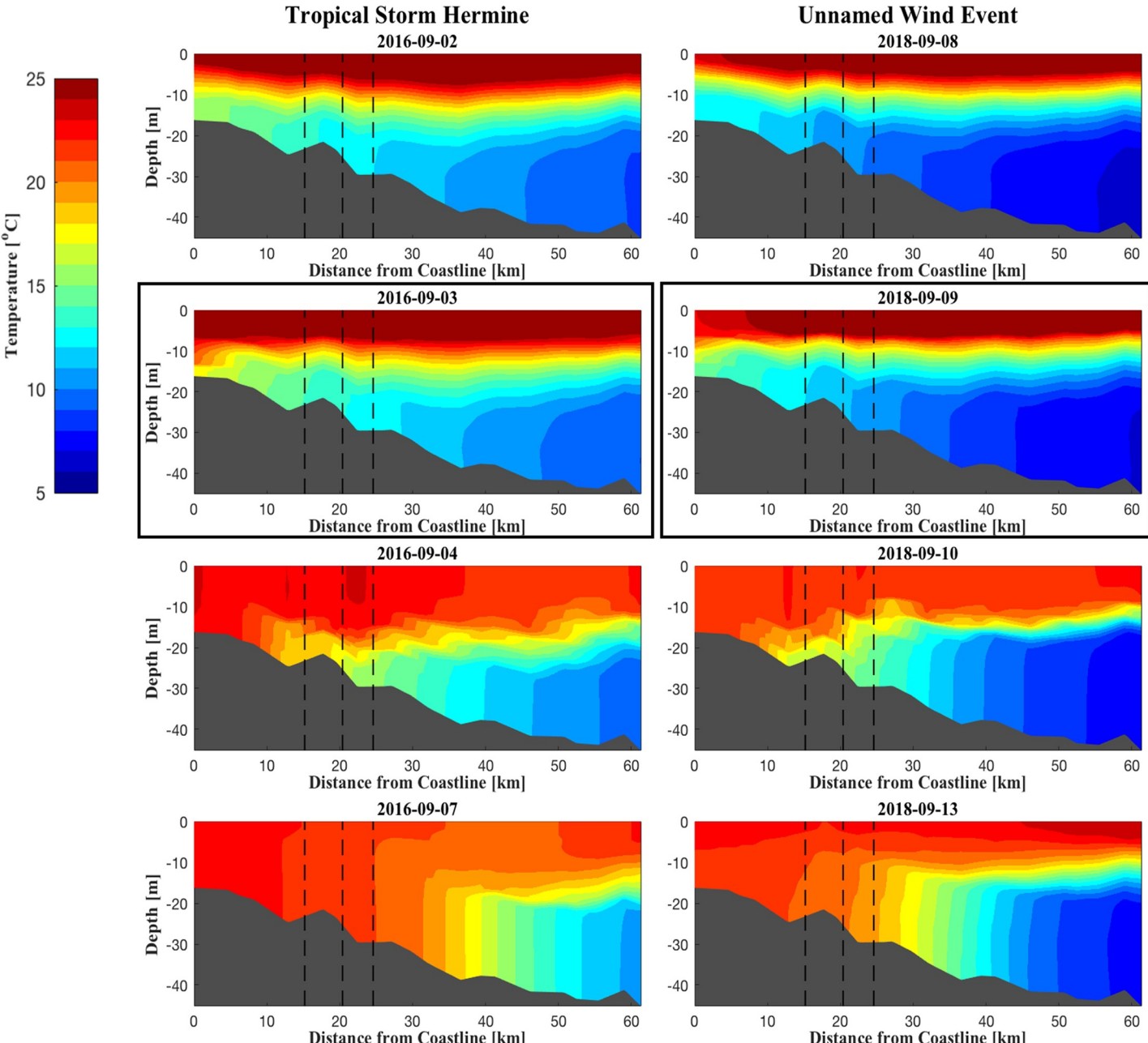

**Fig 5. Modeled bottom water temperature cross-sectional profiles predicted by the FVCOM for storm events, delineated by column, in 2016 and 2018.** * Vertical black dashed lines in each pane refer to the transmitter release locations central to each study site (Southern, Northern, and Middle, for both years in increasing depth and distance from coastline). ** Cross sections are taken along a transect spanning the Middle site (Fig 1), and depict predictions at 00:00 for each given day. *** Panes boxed in black refer to dates of maximum wind speed for the given storm event.

intervention analysis was unable to converge for the time series of transmitters recorded in the Southern site during 2016 and the Northern and Middles site during 2018; inferences related to evacuation were thus not possible for these events (Fig 9). Still, in the case of the 2016 Southern site and the 2018 Middle site, large decreases in the raw time series coincided with storm events.

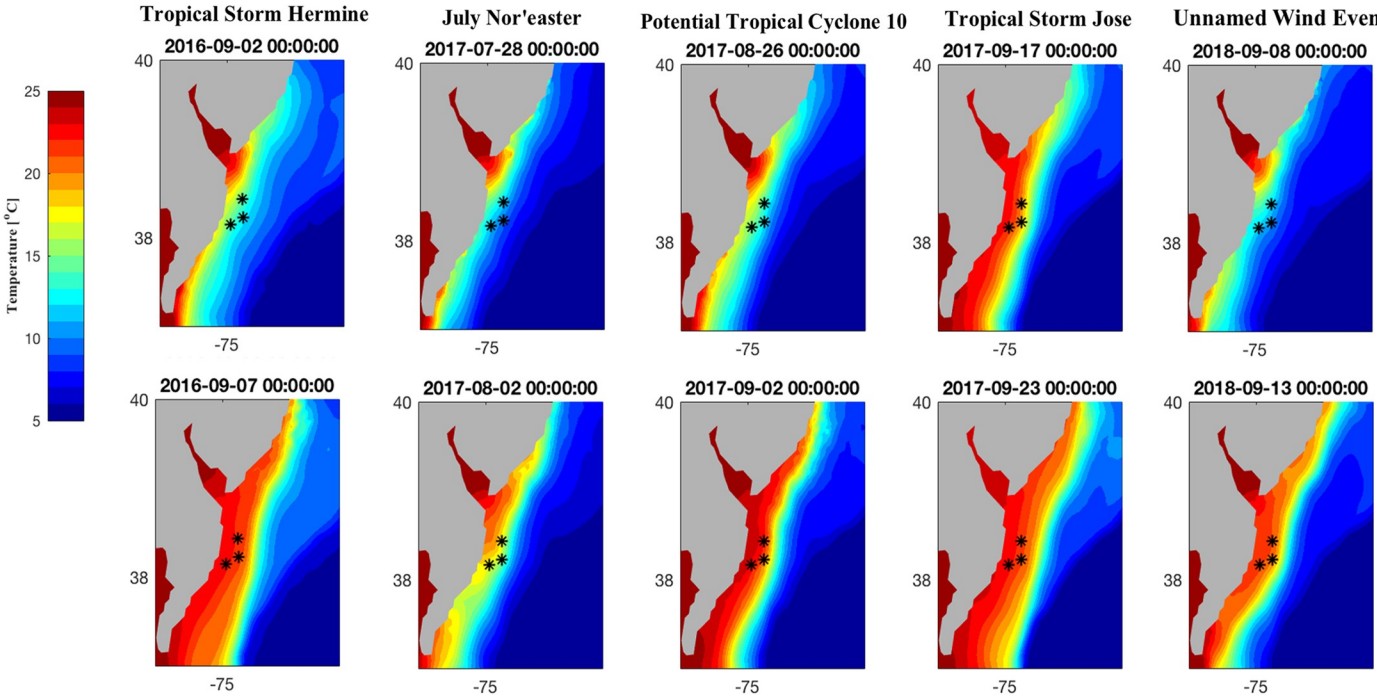

**Fig 6. Modeled bottom water temperature in the southern MAB predicted by the FVCOM for 2016 and 2018 storm events, as well as the first three storm events of 2017, delineated by column.** The fourth storm of 2017 was not included as the thermal structure of the water column did not change following the third storm. * Black asterisks refer to the location of transmitter release, central to each study site.

## Movement analysis: Coupled telemetry-FVCOM mixed effects model

The AIC-selected model indicated that changes in bottom water temperature had the greatest and most significant negative impact on movement index ($\hat{\beta}$ = -0.217; p <0.001), but, contrary to expectations the model failed to detect a significant effect of consecutive cumulative storm impacts, ANSD ($\hat{\beta}$ = 0.002; p = 0.84) (Table 5). Modeled TKE ($\hat{\beta}$ = -0.168; p<0.001) was also influential in the model, with modeled current velocity showing a modest influence ($\hat{\beta}$ = -0.099; p = 0.0014). A significant negative effect was also identified for fish length ($\hat{\beta}$ = -0.19; p<0.001). Both males ($\hat{\beta}$ = 0.617; p <0.001) and individuals of unidentified sex ($\hat{\beta}$ = 0.683; p<0.001) were predicted by the model to have higher movement indices than females. While the presence and magnitude of interactions between sex and size were not directly tested, the distributions of movement indices across tagged individuals suggested an interaction where all small individuals had similar movement rates, while bigger males moved more than bigger females (S9 Fig). The selection of sublegal individuals for tagging likely introduced a bias of a higher proportion of females to males, based on the influence of size on sex change for the species [44]. Still, an overall negative effect of length on movement held across the entire sample of tagged individuals included in the model.

Temporal autocorrelation of the response variable was modeled using the autoregressive terms with lags of 1 and 2 days. These components were incorporated as additional numerical predictors, and both were found to be statistically significant (Table 5).

Movement index significantly differed between sites (ANOVA; p<0.001), with the Middle site exhibiting lower movement indices than the other sites (Tukey contrasts; p<0.001). Movement indices did not differ significantly between the Northern and Southern sites (p = 0.97).

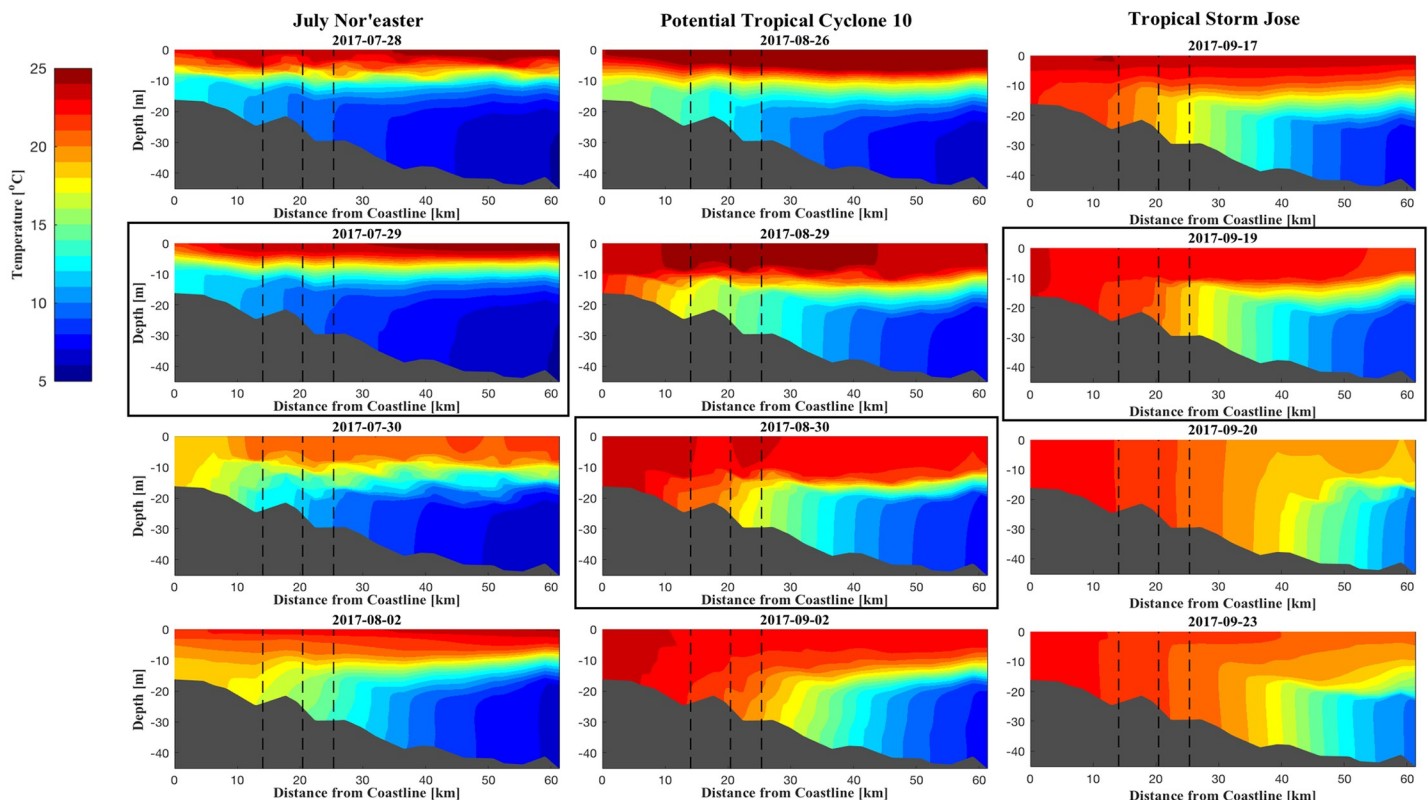

**Fig 7. Modeled bottom water temperature cross-sectional profiles predicted by the FVCOM for the first three storm events in 2017, delineated by column (see Table 3).** * Vertical black dashed lines in each pane refer to the transmitter release locations central to each study site (Southern, Northern, and Middle, increasing depth and distance from coastline). ** Cross sections are taken along a transect spanning the Middle site (Fig 1), and depict predictions at 00:00 for each given day. *** Panes boxed in black refer to dates of maximum wind speed for the given storm event.

## Discussion

By coupling fine-scale telemetry and oceanography, this study demonstrated that storm disturbance was a key driver of seasonal movement behaviors by black sea bass in the shelf waters of the MAB during the late summer and early fall. This study's results indicate that summer-fall storms observed in 2016–2018 varied in intensity, duration, and timing; and had significant, recurring effects on black sea bass habitat conditions that caused large changes in their movement ecology. The initial hypotheses that storms impact black sea bass habitat through rapid

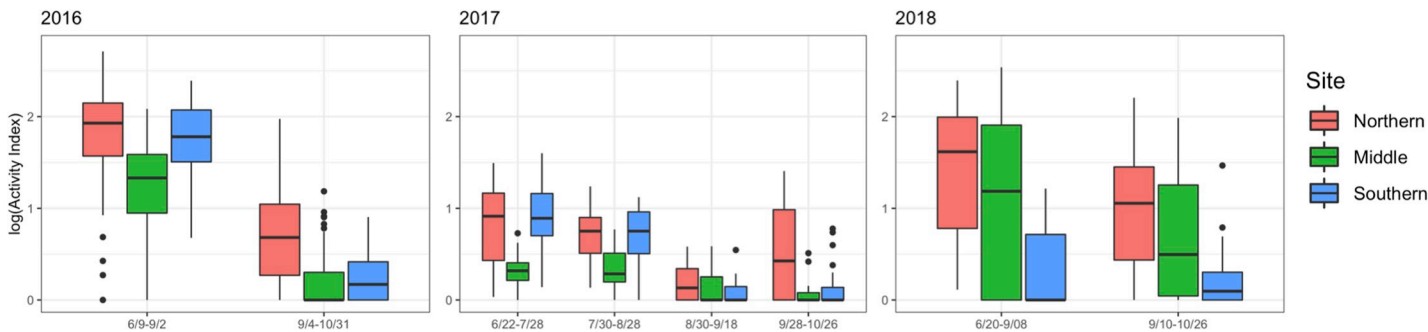

**Fig 8. Hourly black sea bass activity index across study sites for each year.** Boxes are grouped to show activity before and after storm events noted in Table 3.

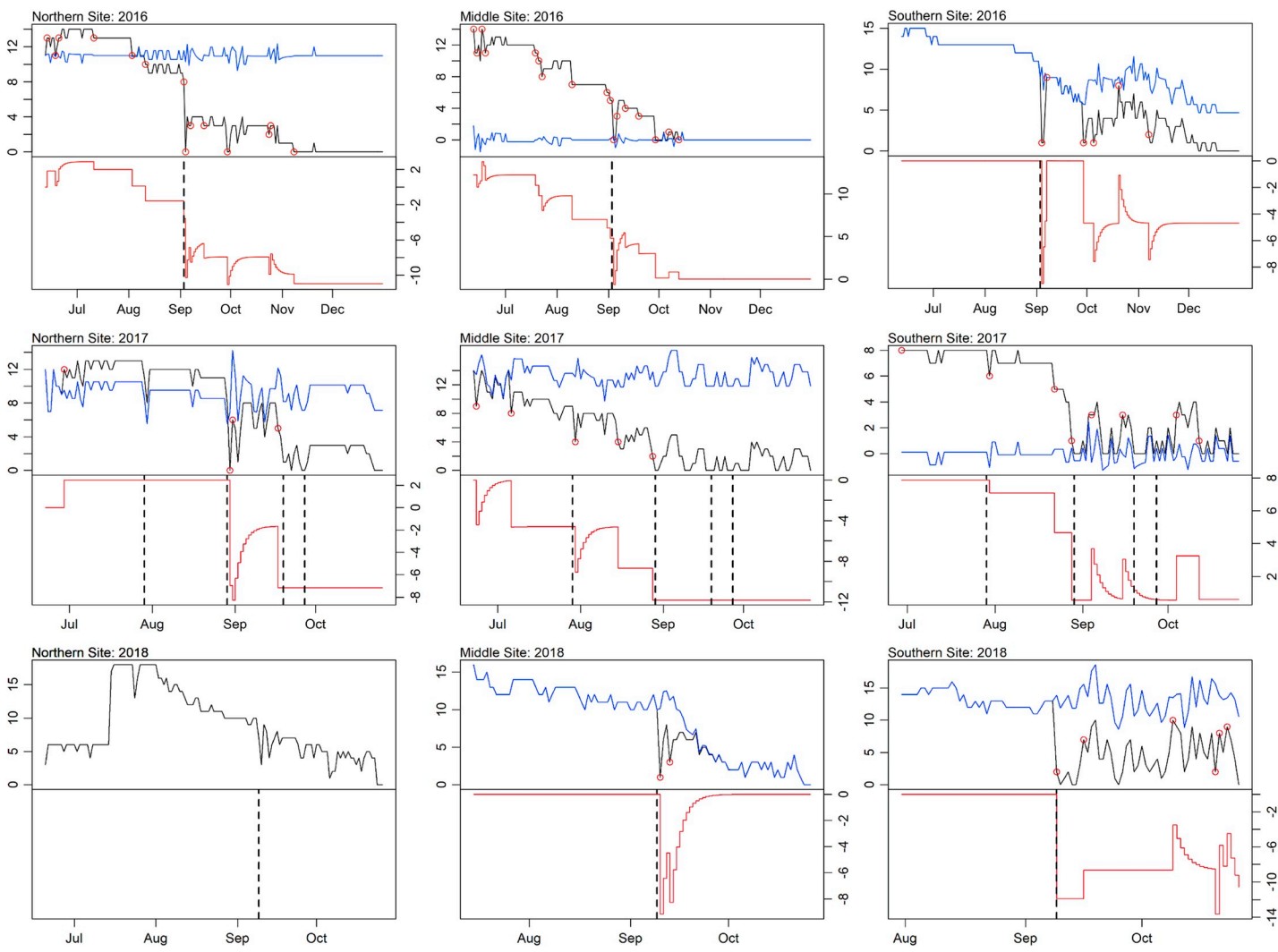

**Fig 9. ARIMA intervention analysis for 2016–2018 study sites.** * The black and blue lines in the upper pane refer to the observed time series of number of fish recorded and modeled ARIMA time series with the influence of identified interventions removed, respectively. The solid red lines in the lower panes depict the influence of identified interventions, where step-wise declines indicate permanent level shifts (true loss) and sharp curvatures that dip and recover indicate temporary shifts (false loss). Location of permanent level and temporary shifts in the fish detection time series are given by small red circles. The vertical dashed black lines refer to the date of maximum wind speed associated with storm events.

changes in temperature, current velocity, and turbulent kinetic energy were supported; furthermore, these storm-driven environmental changes were associated with decreases in movement behavior. In a multi-storm year, 2017, we failed to detect a relationship between cumulative consecutive storm days (ANSD) and depressed movements, but rather observed that depressed movement occurred as a threshold response to a late-season storm, similar to what occurred in other study years. This effect of late-season storm disturbance occurred across all the three sites and resulted in an approximately 50% decreased activity level, which in most instances was also associated with incomplete evacuations. Movement patterns also covaried with length and sex, with males exhibiting higher movement levels (independent to storm effects), a result previously reported in the New York Bight by Fabrizio et al. [40].

Local- and broad-scale movement behavior depended on the timing of seasonal storms and the relative stability of the cold pool. Mid-summer storms (such as the July nor'easter of 2017)

**Table 4. Black sea bass transmitter loss rates.**

| | | Transmitter loss rates | | |
|---|---|---|---|---|
| | | Instantaneous loss rate | | Relative loss rate (%/day) |
| Year | Site | Slope | R$^2$ | |
| 2016 | Northern | -0.013 | 0.811 | 1.196 |
| | Middle | -0.021 | 0.822 | 2.093 |
| | Southern | -0.005 | 0.870 | 0.490 |
| 2017 | Northern | -0.015 | 0.771 | 1.331 |
| | Middle | -0.015 | 0.965 | 1.390 |
| | Southern | -0.021 | 0.793 | 1.198 |
| 2018 | Northern | -0.011 | 0.919 | 1.089 |
| | Middle | -0.012 | 0.773 | 1.178 |
| | Southern | -0.004 | 0.843 | 0.453 |

* Relative loss rate was calculated as the average percent decrease in transmitter presence per day.

did not incur permanent destratification, nor were they associated with changes in movement metrics or evacuations across sites. Storms that occurred later in the year (TS Hermine in 2016, PTC10 in 2017, and the unnamed wind event in 2018), however, triggered permanent break-downs of the cold pool and destratification of the water column in waters less than 30 meters. As such, these storms caused significant declines in activity levels as well as higher numbers of fish evacuating reef habitats across sites. The mechanism driving the impact of these late-season storms on stratification—and subsequently fish movement—was not identified during this study; however higher degrees of surface heating, and thus higher magnitudes of temperature difference at the surface vs. base of water column might occur during late summer and early fall, preconditioning cold pool destruction through storm-driven turnover [24, 25].

Evacuations are a faunal response to catastrophic environmental change [8, 59–61], and, although their occurrence overlapped with the fall migration period, they also occurred in each year of our study during and immediately after storm events. Furthermore, analysis of storm-driven environmental variables during the multiple-storm year, 2017, indicated storm-driven destratification was the primary catalyst for evacuations. In 2017, when multiple storm disturbances occurred, no evacuations were observed during storms that followed the ultimate

**Table 5. Results from the GAMLSS for black sea bass movements.**

| Model term | Coefficient estimate ($\hat{\beta}$) | Standard error | t-statistic | p value |
|---|---|---|---|---|
| Intercept | -0.4966 | 0.0601 | -8.259 | <0.001 |
| TKE | -0.1680 | 0.0314 | -5.328 | <0.001 |
| Temperature | -0.2173 | 0.0413 | -5.260 | <0.001 |
| Current velocity, differenced | -0.0992 | 0.0309 | -3.208 | 0.0014 |
| ln(Movement index$_{t-1}$) | 0.2110 | 0.0165 | 12.767 | <0.001 |
| ln(Movement index$_{t-2}$) | 0.0780 | 0.0146 | 5.335 | <0.001 |
| ANSD | 0.0016 | 0.0081 | 0.200 | 0.8413 |
| Sex, male | 0.6170 | 0.0692 | 8.913 | <0.001 |
| Sex, unidentified | 0.6832 | 0.0662 | 10.321 | <0.001 |
| Length | -0.1904 | 0.0295 | -6.459 | <0.001 |

* All numerical predictors were standardized (centered and scaled) prior to incorporation in the model, allowing direct comparisons of the estimated coefficients.

destruction of the cold pool. The number of evacuations across sites peaked with PTC10 in August (when permanent destratification occurred), with a smaller level of evacuations associated with TS Jose in early September (when destratified bottom water temperatures increased and plateaued). Evacuations were not observed during storm activity following cold pool destratification with the passage of TS Maria, a storm that did not instigate as rapid a change in bottom temperature as did its predecessor. This suggests rapid changes in temperature associated with destratification is the major driver of changes in movement behavior. The observed patterns in evacuation rates and the 2017 ARIMA intervention analysis complement the results of the explanatory GAMLSS analysis, which also identified temperature as the dominant variable negatively impacting local movements.

Site differences in water column stability were apparent and related to depth and proximity to the cold pool front, similar to findings by Lentz et al. [25], who examined cold pool thermal structure in the MAB over repeated wind stress events. The cold pool front, which separates offshore-stratified water from inshore-mixed water, extends along the continental shelf in waters ranging between 30 m and 100 m deep [18, 19]. The front is bound in the north by the Nantucket Shoals and the southern perimeter of Georges Bank, and meets mixed inner shelf waters at the 40 m isobath; the front is bound in the south towards the mouth of Chesapeake Bay and Cape Hatteras at the 30 m isobath [18, 62]. The Southern site was located in the shallowest waters and on the fringe of the front, and thus showed the highest level of bottom water temperature variance and associated water column instability. The Middle site was located in the deepest waters; the Northern site was also located in deeper water, and showed temperature changes more similar to the Middle site in comparison to the Southern site. In accordance with proximity to the cold pool front, we observed the smallest storm-driven change in bottom water temperatures at the Southern site, and the greatest change at the Middle site. The shallower Southern site experienced greater short-term variability across a smaller range of bottom temperatures, and was more prone to permanent destratification; in contrast, the deeper Middle site experienced fluctuations across a broader range of magnitude in bottom temperature, and did not destratify as quickly.

Our findings suggest that fish inhabiting reefs with more stable temperature dynamics are less likely to change residency time or local movement patterns than fish inhabiting reefs with less stable temperature dynamics. The lowest number of evacuations occurred at the Southern site across all years. We argue that the Southern site demonstrated a less severe temperature gradient and subsequent lower magnitude of destratification than that occurring at the deeper Northern and Middle sites. Similarly, the ANOVA results comparing the multi-storm year (2017) movement indices across sites identified the greatest difference in movement between the deepest Middle vs. slightly shallower Northern and substantially shallower Southern sites. Again, this difference is likely related to the interaction of depth with cold pool presence.

The two behavioral responses measured in this study—evacuations or decreased movement in the face of acute disturbance to habitat conditions—are distinct modes, which may have carryover effects to feeding, reproduction, and predator evasion. Black sea bass occupy small home ranges on structured habitats (0.14–7.4 km$^2$) [40], where they feed on reef and adjacent seabed prey items [38, 63, 64]. Reproduction has been observed to occur primarily during June-September [38] and occurs frequently with spawning intervals for females estimated to be 2.7 to 4.6 days [65]. Thus late-summer and early-fall storms, which caused depressed movements away from structure likely interfere with feeding and courtship, which occurs in regions adjacent to the reef. The alternate behavior, evacuation, likely disrupts mating systems and feeding territories and incurs greater predation risk. Additional research is called for on changed feeding and reproductive states before and after storms, the fate of evacuees, and whether evacuations contribute to the greater fall seasonal migration to deeper shelf habitats.

Only a single other study on fish movements during MAB storm mixing occurs, albeit no inferences on movement behaviors were directly linked to storm effects. Fabrizio et al. [40, 66], during which Hurricane Isabel passed through the receiver array on September 19, 2003. Hurricane Isabel triggered permanent destratification and increased bottom water temperatures (approximately 13˚C in 12 hrs) [67, 68]. Loss rates of black sea bass estimated from data in Fabrizio et al. [67] did not exhibit the same strong episodic losses associated with storm events as we detected. In contrast, summer flounder did show evidence of a particularly strong loss in tagged fish from the site coincident with Isabel. Differences in loss rates between studies may be related to the location and depth of the cold pool off the coast of New Jersey vs. off the coast of Maryland.

Key limitations to our findings related to study design include assumptions that (1) the three-receiver array sufficiently overlapped with the distribution of black sea bass at each reef site; (2) that movement rates were realistically indexed as unique movements between receivers; and (3) that daily synchrony between peaks in storm winds and tag losses were evidence for storm-driven evacuations, while slower decays in tag presence resulted from seasonal departures or predation. The study design did not adjust for site differences in reef dimensions or differential usages as sites of refuge, forage, and reproduction [40, 64, 69]. Where fish were caught and released may have also caused differences in how well their home ranges were represented across sites and years. Importantly, this study did not account for changes in vertical movement behaviors in response to storm disturbances; a strong expectation in the literature is that disturbed reef fishes become more tightly coupled to structure [11–13]. Unpublished data from a 2019 biotelemetry study at the Northern site did indeed show that black sea bass used deeper habitats and showed less vertical movement immediately following an August storm-destratification event (D. Secor, CBL, pers. obs.). Furthermore, despite the episodic losses associated with storm events, this study cannot definitively distinguish evacuation from seasonal emigration into deeper shelf habitat. Additional sources of uncertainty surrounding evacuation behaviors include tag loss unrelated to storm-driven evacuation, such as tag shedding, predation, or capture of tagged individuals by anglers.

Longer-term carryover effects owing to disruptions in activity and site fidelity by storm-induced destratification include alterations in the timing of fall-winter migrations and regional shifts in summer habitats. These consequences are relevant in the context of a changing climate, which predicts an increased frequency of high-energy storm events in the NW Atlantic Ocean [70–73]. An increase in high-energy storms within the MAB, depending on timing and track, could produce changes to the location, timing of destruction, and stability of the cold pool [18, 19]. This in turn could induce alterations to the seasonal timing of offshore migrations of black sea bass and other demersal species in the MAB.

Black sea bass support key commercial and recreational fisheries in the MAB shelf system and in regions slated for aggressive wind energy development. Indeed, black sea bass have been established as a model species for understanding wind energy impacts [74, 75], and as a priority species, wind farm and fishing impacts must be assessed against natural storm disturbances. Wind tower construction and maintenance will occur during periods and in regions influenced by natural storm disturbances. Stresses related to wind tower construction include sound caused by piledriving or vessel operation [75–79], alterations to local electromagnetic fields [80, 81]; or, altered distribution of local benthic and demersal species through the emplacement of additional structured habitat [82–84]. Each of these stresses can interact with storm disturbance, obscuring or enhancing impacts associated with wind tower construction alone. Refuge-seeking behaviors associated with pile-driving or vessel noise may be similar in kind to depressed movements associated with storms. Storm-induced evacuations could obscure departures associated with wind turbine construction. Similarly, reduced catchability

could be erroneously associated with wind turbine impacts following a period of high storm activity.

The nature of the potential interaction of natural storm disturbance with wind farm construction disturbance is not well known, and future research on whether this interaction is beneficial or detrimental to fish abundance in affected regions is critical for future management. Furthermore, collective impacts from wind farm construction coupled with impacts from natural storm disturbance could lead to altered fish abundances in regions along the cold pool front. Anecdotal reports from charter fishers suggest greatly reduced catch rates following major storms (D. Zemeckis, Rutgers University, pers. comm.). Storms may also catalyze fall departures to deeper shelf environments shifting fisheries and influencing their accessibility to bottom trawl surveys [85]. The need for storm disturbance to be incorporated in baseline and impact monitoring is heightened by the prediction of higher occurrence of higher intensity storms related to climate change.

## Supporting information

**S1 Fig. The FVCOM estimates of hourly bottom water temperature (˚C), offshore current velocity (m s$^{-1}$), and turbulent kinetic energy (TKE; m$^2$ s$^{-2}$), averaged for the Northern site for August-September 2016 and June-October 2017–2018.** *Dashed red lines refer to modeled maximum wind speeds occurring during each of the six identified storm events (see Table 3). (TIF)

**S2 Fig. The FVCOM estimates of hourly bottom water temperature (˚C), offshore current velocity (m s$^{-1}$), and turbulent kinetic energy (TKE; m$^2$ s$^{-2}$), averaged for the Southern site for August-September 2016 and June-October 2017–2018.** *Dashed red lines refer to modeled maximum wind speeds occurring during each of the six identified storm events (see Table 3). (TIF)

**S3 Fig. (a-c)** Modeled and observed hourly bottom water temperature values across sites for 2016–2018, respectively. Vertical black dashed lines refer to maximum wind speed dates for identified storm events. (TIF)

**S4 Fig. Modeled current velocity cross-sectional profiles predicted by the FVCOM for storm events in 2016 and 2018.** * Red colors refer to offshore current movement, and blue colors refer to inshore current movement. ** Vertical black dashed lines in each pane refer to the transmitter release locations central to each study site (Southern, Northern, and Middle, for both years in increasing depth and distance from coastline). *** Cross sections are taken along a transect spanning the Middle site, and depict predictions at 00:00 for each given day. **** Panes boxed in black refer to dates of maximum wind speed for the given storm event. (TIF)

**S5 Fig. Modeled current velocity cross-sectional profiles predicted by the FVCOM for storm events in 2017.** * Red colors refer to offshore current movement, and blue colors refer to inshore current movement. ** Vertical black dashed lines in each pane refer to the transmitter release locations central to each study site (Southern, Northern, and Middle, for both years in increasing depth and distance from coastline). *** Cross sections are taken along a transect spanning the Middle site, and depict predictions at 00:00 for each given day. **** Panes boxed in black refer to dates of maximum wind speed for the given storm event. (TIF)

**S6 Fig. Modeled turbulent kinetic energy (TKE) cross-sectional profiles predicted by the FVCOM for storm events in 2016 and 2018.** * Vertical black dashed lines in each pane refer to the transmitter release locations central to each study site (Southern, Northern, and Middle, for both years in increasing depth and distance from coastline). ** Cross sections are taken along a transect spanning the Middle site, and depict predictions at 00:00 for each given day. *** Panes boxed in black refer to dates of maximum wind speed for the given storm event. (TIF)

**S7 Fig. Modeled turbulent kinetic energy (TKE) cross-sectional profiles predicted by the FVCOM for storm events in 2016 and 2018.** * Vertical black dashed lines in each pane refer to the transmitter release locations central to each study site (Southern, Northern, and Middle, for both years in increasing depth and distance from coastline). ** Cross sections are taken along a transect spanning the Middle site, and depict predictions at 00:00 for each given day. *** Panes boxed in black refer to dates of maximum wind speed for the given storm event. (TIF)

**S8 Fig. Modeled bottom water turbulent kinetic energy in the southern MAB predicted by the FVCOM for 2016 and 2018 storm events, as well as the first three storm events of 2017, delineated by column.** * Black asterisks refer to the location of transmitter release, central to each study site. (TIF)

**S9 Fig. Distributions of log-transformed movement index across individual tagged fish in 2017, ordered by increasing length and color-coded by sex; F, M, and U refer to female, male, and unidentified fish, respectively.** X-axis labels refer to the tag number, as well as the length of the individual (mm). (TIF)

## Acknowledgments

We are grateful to Dr. Helen Bailey (Chesapeake Biological Laboratory, University of Maryland Center for Environmental Science) for providing valuable insights and careful review during the development of this manuscript. We are also grateful to Cpt. Dan Stauffer and the crew of the *F/V Fin Chaser* for their cooperation in identifying reef habitats, catching and tagging black sea bass, and deploying and retrieving acoustic receivers. Ella Rothermel, Ben Frey, Reed Brodnik, Carlos Lozano, Nicole Barbour, and Teddy Secor also assisted with tagging fish as well as deploying and recovering receivers. Lastly, we are grateful to Catherine McCall for her insights to the study design of this project.

## Author Contributions

**Conceptualization:** David H. Secor.

**Data curation:** Caroline J. Wiernicki, Michael H. P. O'Brien, Fan Zhang, David H. Secor.

**Formal analysis:** Caroline J. Wiernicki, Michael H. P. O'Brien, Fan Zhang.

**Funding acquisition:** David H. Secor.

**Investigation:** Caroline J. Wiernicki, Michael H. P. O'Brien, Ming Li.

**Methodology:** Michael H. P. O'Brien, Fan Zhang, Vyacheslav Lyubchich, Ming Li.

**Project administration:** Caroline J. Wiernicki, Michael H. P. O'Brien, David H. Secor.

**Resources:** David H. Secor.

**Software:** Michael H. P. O'Brien, Fan Zhang, Ming Li.

**Supervision:** Ming Li, David H. Secor.

**Validation:** David H. Secor.

**Visualization:** Caroline J. Wiernicki, Michael H. P. O'Brien.

**Writing – original draft:** Caroline J. Wiernicki.

**Writing – review & editing:** Michael H. P. O'Brien, Fan Zhang, Vyacheslav Lyubchich, Ming Li, David H. Secor.

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
