## [Decision Letter · Decision Letter 0]

27 May 2020

PONE-D-20-11553

The recurring role of storm disturbance on black sea bass (Centropristis striata) movement behaviors in the Mid-Atlantic Bight.

PLOS ONE

Dear Dr. Wiernicki,

Thank you for submitting your manuscript to PLOS ONE. After careful consideration, we feel that it has merit but does not fully meet PLOS ONE’s publication criteria as it currently stands. Therefore, we invite you to submit a revised version of the manuscript that addresses the points raised during the review process, in particular, the cooments and issues taht Reveiwer #2 highlighted about content and structure. Also, the reviewers also highlight that the paper is not directly relevant for offshore wind or fisheries management, as the methods and results are unrelated to that. So either more detailed information on  offshore wind and fisheries management needs to be included in the paper, or they should not be mentioned at all, or only be mentioned in passing in the introduction without further details. 

We look forward to receiving your revised manuscript.

Kind regards,

Vanesa Magar, Ph.D.

Academic Editor

PLOS ONE

Journal Requirements:

Reviewers' comments:

Reviewer's Responses to Questions

**Comments to the Author**

1. Is the manuscript technically sound, and do the data support the conclusions?

Reviewer #1: Yes

Reviewer #2: Partly

2. Has the statistical analysis been performed appropriately and rigorously? 

Reviewer #1: Yes

Reviewer #2: Yes

3. Have the authors made all data underlying the findings in their manuscript fully available?

Reviewer #1: Yes

Reviewer #2: Yes

4. Is the manuscript presented in an intelligible fashion and written in standard English?

Reviewer #1: Yes

Reviewer #2: No

5. Review Comments to the Author

Reviewer #1: This study investigates the impacts of storm events on black sea bass movement behavior. The authors clearly identify other literature on the topic and how their research relates. The authors hypothesize that (1) storm events are a recurring feature that impact black sea bass habitat through changes in temperature, bottom current velocity, and turbulent kinetic energy; (2) changes in movement behavior are caused by both individual and cumulative storm-driven environmental changes; and (3) storm-related movement behaviors are driven chiefly by rapid (<1 d) mixing and increased bottom temperature. The methodology is appropriate and sufficiently presented to allow experiments to be reproduced. The data demonstrate that storm disturbance is a key driver of seasonal black sea bass movement behavior. Importantly, the authors clearly define the limitations of this study in the discussion. They also make a compelling argument for the importance of this research in the context of both climate change and offshore wind energy development. There was a reference to the importance of this type of research to future fisheries management in the abstract, but this was not elaborated on in the discussion, which could have been useful.

Overall this article is organized, clear, and concisely written. It satisfies all of PLOS ONE’s criteria.

Reviewer #2: Overall comments:

While this study, data, and results are important (and exciting!) for both understanding the impacts of storms on fish and for management of black sea bass, the manuscript needs to be improved substantially. Specifically, major edits are needed for:

1) Distinguishing the difference between low and high storm intensities: much of the literature is focused on the effects of hurricanes or large tropical storms. The MAB region receives many storms throughout the summer and fall months but only 6 storms are recorded in this study. While this is fine, the authors should distinguish between background levels of storms vs. the more severe storms focused on in this study;

2) The physiological effects of storms on fish: authors used minimal evidence to argue negative effects of storms and the links between stressors and physiology are not well developed here. Are the authors just arguing that warmer bottom temperatures cause black sea bass to reduce movement to conserve energy? Much of the issues related to this point can be found in the specific comments;

3) Placing the study into context with variability of fall overturn in this region: these storms are important for fall overturn and their variability likely leads to variability in fall overturn. This is important and should be developed more;

4) Distinguishing between the behavioral responses of decreased movement vs. evacuation: these are very different responses, having varying impacts on the population and should be discussed more. Especially as much of the literature used in this manuscript only focuses on evacuations of fish;

5) Distinguishing which months the authors specify as summer and fall months which likely should reference seasons based on the literature focused on fall overturn in the MAB. The authors refer to many of the storms as summer storms when they are actually occurring during the beginning of fall overturn. This is important because stratification during summer is very strong and summer storms do not typically break up stratification. Fall storms will break down stratification and is already known to be a time of significant change in the system and well documented movements of fish species;

6) Explaining if there were any occurrences of fish returning after a storm event. This has been seen in some papers and would be interesting if this happened to black sea bass, especially during the earlier storms of 2017. If there is no evidence the fish returned or fish returns were not detected, that is fine but should be explained;

7) The text overall should be edited to improve clarity and readability. Some cases of this are specified in my specific comments and edits.

General Comments:

Introduction

I would advise the introduction to be reorganized so that the reader is not switching back and forth between reading about storm impacts and the MAB. Perhaps addressing the storm disturbance and impacts on fish first and then introducing the MAB system, the effects of storms on the MAB and potential impacts on black sea bass would flow better. Also, the introduction is a great place to distinguish which months of the year are classified as summer and fall. Finally, in the description of the MAB, the variability of the fall overturn should be explained, perhaps also in reference to changing phenology of the system, so that the storm events measured can be placed into context of interannual differences in the MAB.

Methods

The authors did a good job explaining the tagging, telemetry, ocean modelling and data analysis. I had a couple of concerns with the statistical analyses. There is no discussion about sample size in the GAMLSS. For example, there was a positive influence of being male on movement behavior (in results section) yet there are only 7 males included in this analysis. There should be some test or analysis to show that the sample sizes used in this model are adequate. Also, for the daily movement index, how does the data maintain independence if the daily movement can include movement of one fish over multiple days? If this is incorrect, then please clarify this in the methods. Besides overall text editing (the authors switch between present and past tense), the rest of my comments are specific, see below.

Results

Once the determination of summer and fall is better described, the authors should edit the respective statements accordingly. This applies to the entire results section.

Discussion

What seems to be an important and exciting result is that it appears the evacuation of black sea bass really only occurs in full when the storms are in the fall (September months) and this is not affected by prior storms in the late summer (in 2017). The author’s mention the hypothesis that Secor et al. pose (fall storms are the trigger for offshore migration) and these results in my opinion add supporting evidence of this. However, this result is never fully developed nor placed into context with the hypothesis above. I suggest revisiting it. Also, the authors need to place their results in context with other fish studies in the end to compare how these results may be different or similar to other study systems. Finally, the last paragraph on offshore wind energy seemed very out of place, and definitely not a good conclusion to the paper. There are many more important aspects that the results of this research can be placed into context of (i.e. climate change and increasing storm frequency, shifting fall phenology in the MAB, etc.). The effect of noise, electromagnetic fields, and prey fields was never analyzed in this paper and the relation to storms and development of offshore wind farms seems out of place and a stretch to make. If this is actually a main theme of the paper, then I highly suggest the authors edit and introduce the impacts of wind energy in the introduction when describing the MAB system. But my suggestion is to not include this information. Wind energy is definitely an important and timely study area, but this study did not really address wind energy in the analyses and the results of this study are important enough to stand without tying them into wind energy.

Specific comments are in attached review document.

6. PLOS authors have the option to publish the peer review history of their article (what does this mean?). If published, this will include your full peer review and any attached files.

Reviewer #1: No

Reviewer #2: No

---

## [Author Response · Author response to Decision Letter 0]

6 Jul 2020

Dear Dr. Magar:

Thank you for your very thorough review. We know that this takes a great deal of time and care on your part to ours and the manuscript’s benefit and improvement. We have addressed all general and specific comments below, referencing line numbers for more substantive changes in the clean/marked copy.

Sincerely Yours,

Caroline Wiernicki, NOAA Knauss Fellow 

cc: M O’Brien

 F Zhang

 V Lyubchich

 M Li

 D Secor

Review of Wiernicki et al. THE RECURRING IMPACT OF STORM DISTURBANCE ON BLACK SEA BASS (CENTROPRISTIS STRIATA) MOVEMENT BEHAVIORS IN THE MID-ATLANTIC BIGHT. Submitted to PLoS ONE. Research Article 

General Comments/summary:

Summary: The authors’ aim was to describe and analyze the impact of storms in the MAB in association with the physical breakdown of the cold pool and the effect on fish movement, specifically of black sea bass. This was achieved by using telemetry data and modeling the oceanography of the system using FVCOM. By using data from multiple years, the authors were able to show destratification of the MAB after significant storm events and evidence of changing movement behaviors of black sea bass after storms, which they likened to the rapid increase of bottom temperature and to a lesser degree of increased turbulence and current velocity. By chance, one of the study years (2017) included multiple storm events which allowed the authors to assess the effect of multiple storms on the system. Here they showed earlier storms did not have the same effect as later storms, but did gradually breakdown the cold pool. 

Thanks for this very succinct summary of the paper. 

Major comments: While this study, data, and results are important (and exciting!) for both understanding the impacts of storms on fish and for management of black sea bass, the manuscript needs to be improved substantially. Specifically, major edits are needed for:

1) Distinguishing the difference between low and high storm intensities: much of the literature is focused on the effects of hurricanes or large tropical storms. The MAB region receives many storms throughout the summer and fall months but only 6 storms are recorded in this study. While this is fine, the authors should distinguish between background levels of storms vs. the more severe storms focused on in this study; 

Thanks for this request for greater specificity. We have used the Beaufort Wind Scale and more careful reporting of wind speeds to distinguish higher intensity storm and wind events throughout the paper. Of course, it’s not just about wind speed – pre-existing ocean state, direction, and other aspects will impact the intensity of a given storm, but we do agree some criteria are needed here.

2) The physiological effects of storms on fish: authors used minimal evidence to argue negative effects of storms and the links between stressors and physiology are not well developed here. Are the authors just arguing that warmer bottom temperatures cause black sea bass to reduce movement to conserve energy? Much of the issues related to this point can be found in the specific comments; 

A valid point. We have addressed this below, but were aided by discovering a recent paper specific to black sea bass that allowed us to estimate reduced aerobic scope associated with a change from 12 to 24 C, which was fairly typical for destratification events. Other more general literature has also been added on likely physiological effects.

3) Placing the study into context with variability of fall overturn in this region: these storms are important for fall overturn and their variability likely leads to variability in fall overturn. This is important and should be developed more; 

We agree that the role of storms in driving fall overturn in this region is important. However, while the three-year stimulation discussed in our study has shed some light on how storms affect the fall transition process in the water column from highly stratified to well mixed, we believe it is far from sufficient to explain the interannual variability of fall overturn in this region. For inter-annual variability studies, at least a decade-long simulation is typically needed to provide a reliable assessment of how different storm characteristics affect the timing of fall overturn. We believe this is beyond the scope of this paper.

4) Distinguishing between the behavioral responses of decreased movement vs. evacuation: these are very different responses, having varying impacts on the population and should be discussed more. Especially as much of the literature used in this manuscript only focuses on evacuations of fish;

We have sought to make this distinction between these two behavioral responses, as well as their implications, greater throughout the manuscript and primarily in the discussion. The binary response of some individuals to evacuate vs. some to hunker down likely reflects some biological or physiological distinction that in turn influences behavior. We see some support for this in the characteristics of the tagged fish whose movement declines were influenced by sex and length. There are likely carryover effects in population structure based on who stays and who goes.

5) Distinguishing which months the authors specify as summer and fall months which likely should reference seasons based on the literature focused on fall overturn in the MAB. The authors refer to many of the storms as summer storms when they are actually occurring during the beginning of fall overturn. This is important because stratification during summer is very strong and summer storms do not typically break up stratification. Fall storms will break down stratification and is already known to be a time of significant change in the system and well documented movements of fish species; 

The study period corresponds to the transitional period between summer and fall, and subsequently seasonal influences are not so easily attributed to one or the other. From a temperature point of view September is summer, but we realize this is unconventional. We have addressed this by avoiding seasonal terms in places and where we reference seasons, use conventional month intervals. 

6) Explaining if there were any occurrences of fish returning after a storm event. This has been seen in some papers and would be interesting if this happened to black sea bass, especially during the earlier storms of 2017. If there is no evidence the fish returned or fish returns were not detected, that is fine but should be explained; 

Yes, we did see instances of return following several days to several weeks’ time. We now report on these, but please note evacuations are analyzed by ARIMA and operationally defined as permanent evacuations. Another related aspect that in 2017, two fish returned to the same site where they were initially tagged the previous summer (Secor et al. 2019). 

7) The text overall should be edited to improve clarity and readability. Some cases of this are specified in my specific comments and edits. 

We are grateful for the attention paid to the readability and flow of the text. Adjustments were made throughout the document to improve its clarity, both in direct response to reviewer comments and elsewhere.

Thank you once again for the attention our manuscript. We have endeavored to match this in our revisions. For brevity, our responses to specific comments are fairly direct.

Author response to specific comments/edits on text, tables, and figures:

Title

The phrase “recurring role of storm disturbance” in the title is ambiguous. Are the authors referring to recurring storms and their impact or that there is continued evidence of impacts of storms on black sea bass? 

 - Line 3: Title was changed to The recurring impact of storm disturbance on black sea bass (Centropristis striata) movement behaviors in the Mid-Atlantic Bight

Abstract 

Line 25: Summer storms do not cause rapid destratification in the MAB. Fall storms yes, but not summer. Please revisit classification of what is considered a summer or fall storm. I have multiple comments about this in the rest of the specific comments. 

 - Line 24-25: “Summer” was removed in the initial description of storm events, and events were clarified as occurring during the late summer and early fall.

Line 28: “model ubiquitous demersal” – while black sea bass relatively stationary behavior makes them ideal for telemetry studies, it is not clear why the reader should care about depressed movements of an already stationary fish. Perhaps rephrase to emphasize “model” organism relates to their use in tagging studies but not as much on the potential impact of storms. 

 - Lines 27-30: Sentence was restructured to emphasize black sea bass as a model species for the ease of which they can be tagged and also in how their sedentary, demersal life history exposes theme regularly to storm disturbance. 

Line 33: is this 8-15 black sea bass per year or over the entire 2016-2018 time frame? Please specify. 

 - Line 33: Clarification was included that 8-15 black sea bass were released “each year”.

Line 33: “at each of three reef sites” – this is confusing as the authors have not introduced the three reef sites yet in the abstract. I suggest rephrasing this statement. 

 - Line 33: “each” was removed so as to avoid confusion regarding the reef sites.

Lines 34-35: “…activity levels and reef departures of black sea bass, and fluctuations in modeled temperature …”

 - Lines 34-36: Sentence was formatted following reviewer suggestion: “Data were analyzed for activity levels and reef departures of black sea bass, and fluctuations in modeled temperature, current velocity, and turbulent kinetic energy. 

Lines 36-37: How do the authors reconcile differing behaviors of low activity vs. evacuation? Is this an artifact of the methodology or truly binary responses to the storm events? How do the authors know if late season departures are not associated with offshore migration?

 - Lines 36-37: Methodology is discussed in greater detail further in the paper; no adjustments made here in the abstract other than to specify that not all fish evacuated. 

Lines 40-42: The conclusion of the abstract feels like a stretch. Where are anthropogenic impacts measured in this study? What are the anthropogenic impacts (i.e. fishing? Climate change? Pollution?). Perhaps a stronger conclusion is to place these results in context with climate change and the predicted increase of storm frequency and intensity along the MAB in the future. 

 - Lines 41-44: Conclusion was edited to emphasize how black sea bass assessments of fishing and wind farm development impacts should be performed in the context of storm disturbance. 

Introduction

- General: Introduction was restructured to reflect the following: first describing the impacts of storms as a disturbance on fish; second describing the MAB and how storms may disturb the region; third, discussing potential impacts on black sea bass in the MAB and leading into hypotheses.

Line 50: “less physical structure” – This is intriguing but what might be the potential difference or concern between a region with varying degrees of physical structure?; “subject to hurricane-forcing” – this statement feels out of place because some of the other regions referenced above are not subject to hurricane forcing. Is this insinuating that we are only concerned about regions with hurricane-forcing and those with lesser storms are not as important? *This is an area where the authors could develop their ideas further. For example, what makes the MAB unique compared to these other systems is the cold pool and the ability for it to breakdown during storms. Placing this into context with the other systems referenced, especially in relation to rapid temperature changes, would provide a stronger argument to why the MAB system is important to study.

 - Line 51-53: Rephrased to shift emphasis on presumption that deeper waters would be less impacted by storms and that such systems are difficult to study. 

Line 87: Rewrite as - “Thus, the literature supports that severe storms can disrupt demersal fish communities”. – along these lines, the examples from the literature only provide evidence of fish evacuations not reduced movement behavior…is there evidence for this in other fish species? It would be useful to provide a variety of responses of fish to storms especially because the authors found depressed movement. If this study is the first to show depressed movement, then this should be explained later in the discussion. 

 - Lines 63-64: Examples of severe storms driving depressed movement in reef fish provided.

 - Line 64: Inclusion of term “severe”.

Line 87-89: “Still, the concept … been fully explored” – this statement needs to be developed more. Do all studies above represent singular extreme events? Why is it important to study if storms occur each year in comparison to the other studies? 

 - Lines 66-71: Additional description of the unexplored potential for storms to serve as a disturbance regime.

Line 53: “…Cape Hatteras, North Carolina,…”

 - Line 73: “North Carolina” included to specify location of Cape Hatteras.

Line 58: “…MAB is also vulnerable to storm-driven temperature disturbances …”

 - Line 79: “temperature” added to clarify the nature of the storm-driven disturbance.

Line 59: Remove “overlapping”. 

 - Line 80: “overlapping” removed.

Line 61: “…at the Nantucket Shoals.”

 - Line 81: “the” added prior to “Nantucket Shoals”.

Line 64: References 13-16 mostly focus on fall overturn. This is fine but the seasonality of the cold pool destruction should be clearly stated. If there is more evidence of summer storms impacting the cold pool (besides this study), I would also add those and again be clear about what part of the year this is occurring.

 - Lines 83-86: Terms included to directly identify the seasonality of storm-driven overturn, beginning in the late-summer (mid-late August) and peaking in the fall (September-October). Additional reference including documenting overturn related to a late August storm.

Line 66: How common are nor’easters in the summer? 

 - Line 87: Terms “and fall storm events” added for completion; term “nor’easter” removed.

Line 68: “summer storms” – again, this should be clarified in terms of what months “summer” refers to

 - Lines 90-91: Seasonal definitions of summer and fall included.

Line 69: “rapid partial destratification” – partial vs. full destratification should be explained 

 - Lines 92-94: Definitions of partial and total destratification included. 

Line 70: The values of temperature change are based on a hurricane event. This should be clarified as an extreme event in the system. 

 - Line 89: Specification of the 10 degree increase in bottom water temperature as related to a single extreme event is provided.

Lines 70-71: Refs 18 is focused on juvenile black sea bass not adults and also gradually increased temperatures after acclimation; and refs 19-20 are on temperature tolerance not so much acclimation and very acute exposures. The impact of rapid temperature change on fish physiology is extremely important but this section needs work. Can the authors find studies of acute vs chronic exposure of increasing temperature? Studies on heat shock proteins? Also, there needs to be a link between how the physiological stress of increased temperature will affect fish behavior (i.e. movement). Is there a threshold where a fish cannot move and will not evacuate? Did any of the other papers on fish responses to storms report physiological stress? Here is also where there should be some distinguishing between behaviors of avoidance or mitigation? For example, are the fish leaving the system before the storm or compensating during and after? 

 - Lines 96-104: Greater specifications on behavioral and physiological descriptions of acute and chronic heat responses across a variety of fish species provided. Also included are results form a recent publication specific to black sea bass that shows diminished metabolic scope as temperatures increase from 12 to 24 C.

Lines 73-75: I am intrigued about potential issues related to changes in current velocity, TKE and noise but the authors do not provide further evidence that these are stressors to fish, and as such do not provide a potential hypothesis as to how these physical effects from storms could affect fish. 

 - Lines 107-112: Greater specification and citation of storm generated flow disturbing fish movement, and the potential for storm-generated noise to act as a stressor to fish communities.

Lines 75-76: “Because several storm systems …destratification each year” – this statement is awkward, rewrite. 

 - Lines 112-116: Original sentence broken into two clauses for increased clarity/readability.

Line 77: “disturbance regime” – this seems to be an important concept… the authors should explain this further and show an example of other disturbance regimes to provide evidence to why this should be unique on the MAB

 - Lines 114-115: Additional description of disturbance regime as “seasonal fluctuations in temperature and flow” included.

Line 92: “range behaviors” – this is awkward, please rephrase. 

 - Line 119-120: “range behavior” removed and replaced with “an affinity for both natural and artificial structured habitats”.

Lines 92-94: Re-write as “…centered on artificial and natural structure [25] and are mostly sedentary and reef-associated [26-29], making them amenable to biotelemetry studies on their movement behaviors.” 

 - Lines 120-121: Rephrased: “which makes them an ideal candidate for biotelemetry studies on potential shifts to their movement behaviors”.

Lines 94-97: “Black sea bass and … late October [26,27,30]” – this sentence is awkward, please rewrite this sentence 

 - Lines 121-124: Sentence broken into two clauses for clarity.

Line 95: “…occur inshore throughout the spring, summer and early fall months,…”

 - Line 123-124: Clauses rephrased.

Lines 96-100: So here the authors explain black sea bass movements in relation to their fall migration…so are the storms analyzed in this paper not fall storms? Again, clarifying summer and fall months early on in the paper should help reduce confusion surrounding this issue. 

 - Lines 127-128: Arriving storms clarified as occurring in late summer and early fall.

Lines 99-100: The hypothesis from ref 17 ends the paragraph without any additional follow up from the authors. Is this also hypothesized for this paper? If not, how are the hypotheses and results of this paper placed into context with this hypothesis? 

 - Lines 127-129: Clause added to provide context for the hypothesis in the previous sentence: “The potential role of late summer-fall storm disturbance in MAB to serve as a migratory cue emphasizes the need to understand repeated storms as an ecologically- significant disturbance regime.” 

Line 101: Replace “is” with “was” 

 - Line 130: “is” replaced with “was”.

Line 103: Remove “More specifically” and capitalize “We..”

 - Line 132: “More specifically” removed; “we” capitalized.

Lines 103-108: Are there any hypotheses related to differences in movement behavior (i.e. depressed movement vs. evacuation)? 

 - Lines 132-136: Left unchanged, as hypothesis-specific delineations between evacuations and movements are described further in Methods section.

Materials and Methods

General comments: Section reviewed and corrected for consistent verb tense.

Line 115: Rewrite: “This project included three study reef sites located …”

 - Line 144: Rewritten as: “This project included three study reefs located…”

Line 116: “…Ocean City, Maryland, USA, …”

 - Line 145: “USA” included.

Line 118: “were exposed to” – rephrase this statement; exposed to sinuates some experimental manipulation not that these sites regularly are part of the cold pool 

 - Line 146: Rewritten as: “All study reefs overlapped with the presence of the cold pool…”

Line 147: Was the tank a flow-through tank? If so, this should be mentioned. If not, were there airstones? 

 -Lines 173-174: Sentence “Water within the tank was partially replaced at approximately 10-minute intervals to avoid deoxygenation.” included for clarification on oxygen replenishment of holding tank.

Line 151: After anesthetizing, the authors should include weighing, measuring and sexing the fish. How the authors sexed the fish is very important because they do not explain how they arrived to Unknown statuses for some of the fish in the study and this is important, especially for black sea bass. 

 - Lines 178-181: Additional text included to describe the weighing, measuring, and sexing of fish prior to surgery. 

Lines 153-154: “…was made cranial to the vent, and lateral to the midline.” This phrase is confusing, please rewrite. 

 - Line 184: Rewritten for clarity of incision location: “a 1-cm incision was made lateral to the midline, preceding the vent”.

Lines 157-159: Did tagging the fish not help reduce some barotrauma? If the fish were fully recovered in the tanks post tagging, then how did some of them need help equalizing in the water? Were the fish vented at all? Did the fish get external tags so that if they were accidentally caught fishermen could notify the researchers? 

 - Lines 187-188: Statement on barotrauma observations and procedures included for clarity: “Incisions alleviated internal pressure although barotrauma symptoms were likely not fully abated.” We included a relevant paper on barotrauma in black sea bass and its symptoms. We did not add additional stress by placement of external tags, but rather tagged sublegal fish to reduce fishing takes, described earlier in the same paragraph.

Line 166: Change “Mid-Atlantic Bight” to MAB

 - Line 199: “Mid-Atlantic Bight” replaced with “MAB”.

Lines 170-171: Are there citations for the ROMS ESPreSSO model? 

 - Line 204: Citation for ROMS provided.

Lines 177-178: The authors mention potential stress from current velocity but there is not cited evidence of this for fish nor is this idea developed in the introduction. Also what do the authors mean by “station-keeping”?

 - Lines 215-218: Additional discussion of the potential effects of current velocity included in introduction, with references included again here; “station-keeping” replaced with “maintaining position at reef habitat home ranges”.

Line 185: “Delmarva” – this needs to be explained; most readers not from the US East Coast will not know what Delmarva means 

 - Lines 225-226: “Delmarva” replaced with “Delaware-Maryland-Virginia”.

Line 187: How exactly were the model results evaluated with the observed bottom water temperature? See my comments for figure SI3a-c.

 - Line 227: Phrase “evaluated through comparisons with” replaced with “compared with”.

Line 191: How and where did the authors measure peak wind speed? 

 - Lines 219-222: Description of how observed wind speeds were obtained.

Line 193: How was the threshold of >5m/s chosen? This seems to have filtered out less severe summer and fall storms that occur in the region. In my opinion this is fine, but the presence of only severe storms in the analysis should be explained here and in the discussion. 

 - Lines 234-237: Rationale for the 5 m s-1 wind speed cutoff provided: “The lower limit of 5 m s-1 was selected to provide a conservative definition for potentially disruptive storm activity (3 or greater on the Beaufort Wind Scale [49]). Storms above this limit were further categorized and compared according to the Beaufort Wind Scale.”

Line 195: Change “evens” to “events”. 

 - Line 238: “evens” replaced with “events”.

Lines 216-217: “This approach allows discrimination … caused by storms.” – How is this interpreted to distinguish between fish leaving and coming back vs. false detections of a fish not actually leaving? 

 - Lines 259-269: Section of paragraph rewritten for clarity: “ARIMA intervention analysis facilitates the identification of false evacuations, identified as single points within the recorded time series that temporarily, but significantly, alter the behavior of the rest of the series.” … “The analysis tested for the presence of two types of interventions: (1) temporary shifts and (2) permanent level shifts. Permanent level shifts (stepped declines) are indicative of fish evacuation—interventions that fundamentally and permanently changed the remaining time series. Temporary shifts are those interventions that altered the time series temporarily and appear as nonlinear returns to the previous detection level (see Figure 9).”

Lines 233-235: Did the authors only run one model of 2017? If so, why did they not run models for 2016 and 2018 and remove the ANSD term? This sentence is a bit ambiguous to the reader. 

 - Lines 274-276: Sentence rewritten for clarity. Note that discussion of 2017 as an optimal modeling year based on the repeated occurrence of storm events provided in existing text.

Line 259: How were the different models constructed, considering the authors picked the best fit model based on AIC scores? 

 - Lines 292-294: Moved statement on AIC selection and incorporated additional criteria for how compared models were constructed. 

Results

Line 280: Rephrase to clarify that the storms occurred between the measured time period of June to October (because the storms did not occur in June or October) 

 - Line 310-312: Rephrased clarify that storms occurred during measure period of time June-October.

Lines 282-286: The storms in September to me are early fall storms and the only true summer storm was the one at the very end of July; the August storm is also pretty late for summer. So the storms assessed are really more focused on late summer-fall storms and not “summer storms”. But if during editing this is clarified prior, then please disregard this comment. 

 - Lines 312-317: Text incorporated throughout Introduction and Methods sections to clarify that storms are occurring the during the late summer and early fall.

Line 301: The relatively stable range of 12.5-16.9�C is still a significant change in bottom temperature (and significant for biology). 

 - Line 333-334: This observation now noted in terms of gradual rise in August; sentence also has been simplified.

Line 305: What values are in the parentheses? Mean and SD? If so, this should be clarified. 

 - Line 336: Parenthesis values clarified as mean and standard deviation.

Line 306: Rewrite – “…a destratification event at the end of July,…”

 - Line 338: Rewritten “event at the end of July”.

Line 314: “stronger storms and more moderate storms”. This seems subjective; was there a way to classify storms as “strong” vs. “moderate”?

 - Line 346: Strength of storms now additionally categorized according to Beaufort Wind Scale.

Line 322: In looking at Figure 3, the winds appear to be northeasterly not northwesterly. 

 - Line 356: Corrected to match patterns in Figure 3 of “northeasterly” winds.

Line 381: Replace “all” with “each”

 - Line 420: “all sites” replaced with “each site”.

Line 384: Replace “lowered” with “decreased”

 - Line 421: “lowered” replaced with “decreased”.

Line 385: Replace “than during” with “compared to”

 - Line 422: “than during” replaced with “compared to”.

Line 385: What pairwise comparisons were significant? Is this between site? Or between years? 

 - No adjustment made; text indicates significance across years between given storm events at Lines 423-425.

Lines 385-386: I am confused why the results for 2017 are only in reference to PCT10. Did the authors group together data before and after PCT10? What about the other storm systems? Again, for the Tukey contrasts, is this across sites? It would be nice to know which sites were significantly different from each other. 

 - Lines 424-425: Text added: “with no other significant differences detected before and after the other storm events that year.”

Lines 394-396: Does this mean that when there are more frequent storms there is more evacuation from the shallow site compared to when there are less frequent storms? 

 - No adjustment made; text confirms this within Lines 452-456.

Lines 396-398: I am confused. The authors state that the instantaneous loss rates at the Middle site were the highest during all years except for 2017 and mention that this occurred when the Middle and Northern evacuation rates were similar, but really this was when the Southern site evacuation rate was higher (i.e. Middle and Northern were similar but NOT higher than Southern). 

 - Lines 430-432: Comparison between sites and years is not too important to the paper’s thesis, but rather the episodic loss rates owing to storms – the point here is to quantify baseline loss rates, which were similar across sits. This section has been reduced. 

Lines 397-398: For readers unfamiliar with telemetry studies, it would be nice to put into context the difference in instantaneous loss rates across the years. There are only slight increases in rates; is this still significant? Or does it mean that there is not much interannual variability in evacuation rates all things considered? 

 - Line 430-432: Discussion of comparisons and role of loss rates streamlined; calculations were also re-evaluated and corrected accordingly.

Line 437: Is this from the results of the best fit model? If the authors went through model selection it would be nice to have a clear indication of which model performed the best 

 - Lines 473-476: Specification that the model discussed is the best fit model (lowest AIC; Table 5).

Line 439: So along the lines of my comment above, the best fit model still included the non-significant ANSD term? 

 - Yes, with mention of best fit (lowest AIC) model containing ANSD in Lines 473-476.

Lines 442-443: Remove “although it did not directly relate to storm influence on movement”. This is not quite known and it is obvious there is other biological information included in the model. 

 - Line 478: Removed “although it did not directly relate to storm influence on movement”.

Line 445: If there might be an interaction between sex and length, why was this not included as an interaction? I agree that there likely is an effect of sex on length. If this is not to be used as an interaction, it would be worthwhile for the authors to provide a preliminary test showing there is or is not a relationship between sex and length. 

 - Relationship of sex and length likely present and detected by model, and is discussed further in lines 484-485. Exploration of relationship relative to movement (in light of selection bias of sample for sub-legal individuals and therefore likely females) exceeds scope of study.

Line 449: “…entire sample.” – what do the authors mean by “sample”? I am confused by this statement 

 - Lines 486-487: Added “of tagged individuals”.

Line 460: Remove “the” after exhibiting. 

 - Line 498: “the” removed.

Discussion

Line 472: “…changes in temperature,…”

 - Line 509: “in” added.

Line 483: “Early summer storms” – Not trying to beat a dead horse, but a storm occurring at the end of July does not seem “early” to me.

 - Line 521: “Early summer” replaced with “Mid-summer”.

Line 494-495: It seems misleading to mention that evacuations are an extreme response yet they happened every year in the study without including information that these are also occurring during the time that we expect to see black sea bass start to move offshore. 

 - Lines 533-535: Sentence restructured: Evacuations are a faunal response to catastrophic environmental change [8, 58, 59, 60], and, although their occurrence overlapped with the fall migration period, they also occurred in each year of our study during and immediately after storm events.

Line 504-506: Does this sentence thus indicate that the influence of Maria on water temperature did not instigate as rapid of a temperature change as the other storms? If so, this should be clarified. 

 - Lines 543-545: Text added: “a storm that did not instigate as rapid a change in temperature as its predecessor.”

Lines 516-518 and 521-522: These seem to contradict each other. Do these sentences indicate that the Southern site temperature varies more in frequency than in magnitude when compared to the Middle and Northern sites? If so, please clarify. 

 - Lines 563-567: Text added to clarify relationship of site depth and stability: “The shallower Southern site experienced greater short-term variability across a smaller range of magnitude in bottom temperature, and was more prone to permanent destratification; in contrast, the deeper Middle site experienced fluctuations across a broader range of magnitude in bottom temperature, and did not destratify as quickly”

Lines 525-526: “The lowest rates of transmitter loss and the lowest number of evacuations occurred at the Southern site for all years” – But the authors found higher rates of instantaneous loss rates in the Southern Site in 2017…? I am confused. 

 - Lines 570-571: Calculations of daily transmitter loss rate were re-evaluated and corrected. Section on comparisons and role of transmitter loss rates was reduced.

Line 530: “…relatively shallower Northern” – on the previous page the authors mention that the Northern site is deeper and similar to the Middle site. Please correct and clarify this. 

 - Line 575-576: Text to clarify depth relationships across sites included: “deepest Middle vs. the slightly shallower Northern and substantially shallower”.

Lines 581-584: This paragraph is a good section to include other studies and place the results of this study into context with other fish responses to storms. 

 - Lines 588-590: We would prefer to retain focus on shelf systems where fish are exposed destratification events rather than the mix of storm influences, principally in shallower coastal systems introduced in Introduction. Only the Fabrizio study bears on such storm impacts. We have focused the section to highlight the contrast between these two comparable telemetry studies. 

Lines 538-539: If the results from Fabrizio et al do not have the same results as the results for this study, what potential reasons can explain this? Perhaps this can shed more insight into the results found in this study (i.e. comparing the storm system of that study to the ones in this study). 

 - See above comments

Line 561: Refs 55-57 discuss reduced movement in relation to predatory prey experiments and indirect effects found in communities so I am confused how this can be related to reduced movement from storm impacts (these are very different mechanisms)

 - We accept this point – we were using theoretical literature to support link between movement and foraging habitat, but it was tenuous. Rather, we move directly to how movement relates to ecological functions in black sea bass

Line 565: Ref 27 describes the biology of the South Atlantic stock of black sea bass. Please reference a paper for the northern stock of black sea bass, the species that was used in this study. Also, the September-October is towards the end of the spawning season in this region so I don’t quite know how much of an effect there would be. To me, a more important and concerning aspect is the effect of storm systems on recruitment and the ability for recently spawned gametes and larval fish to ingress during these events. Also occurring at this time is the potential for black sea bass to change sex and this should at least be addressed here as the authors found important differences between the sexes and movement behavior. 

 - Lines 618-619: Reference switched to Drohan et al. 2007, here and elsewhere. Discussion of timing of individuals to change sex briefly discussed earlier in Results (impacts beyond scope of study, as selected a biased pool towards greater proportion of females).

Line 573: “…induce range shifts…” – what do the authors mean by this? In what way do they expect range shifts to occur (i.e. latitudinally? Cross shelf?)

 - Line 627: “latitudinal” included to describe range shifts.

Line 575-588: Refer to my comment in the general comment about this paragraph. 

 Lines 629-648: Similar to the Abstract, this paragraph was changed to first emphasize that black sea bass is a key species for fisheries and understanding wind energy impacts. Greater balance is then given to both of these sectors in the context of storms and increased storminess.

References

Some of the references were not inputted correctly (e.g. Ref 16, 32). Please review all references and make sure the citations are accurate. 

Corrections made to references:

3. Bouchon C, Bouchon-Navaro Y, Max L. Changes in the coastal fish communities following Hurricane Hugo in Guadeloupe Island (French West Indies). Atoll Res Bull. 1994; 422: 1-19.

15. Li Y, Xue H, Bane JM. Air-sea interactions during the passage of a winter storm over the Gulf Stream: A three-dimensional coupled atmosphere-ocean model study. J Geophys Res. 2002; 107(C11): 1-13.

21. Rasmussen LL, Gawarkiewicz G, Owens WB, Lozier MS. Slope water, Gulf Stream, and seasonal influences on southern Mid-Atlantic Bight circulation during the fall-winter transition. J Geophys Res. 2005; 110(C2): 1-16.

24. Beardsley RC, Chapman DC, Brink KH, Ramp SR, Schlitz R. The Nantucket Shoals flux experiment (NSFE79). Part 1: A basic description of the current and temperature variability. J Phys Oceanogr. 1985; 15: 713-748.

25. Lentz S, Shearman K, Anderson S, Pluddemann A, Edson J. Evolution of stratification over the New England shelf during the coastal mixing and optics study, August 1996–June 1997. J Geophys Res. 2003; 108(C1): 1-14.

45. Zhang F, Li M, Ross AC, Lee SB, Zhang D. Sensitivity Analysis of Hurricane Arthur (2014) Storm Surge Forecasts to WRF Physics Parameterizations and Model Configurations. Wea Forecasting. 2017; 32(5): 1745–1764. 

67. Sedberry GR, Van Dolah RF. Demersal fish assemblages associated with hard bottom habitat in the South Atlantic Bight of the U.S.A. Environ Biol Fish. 1988; 11(4): 241–258.

70. Vermaire JC, Pisaric MFJ, Thienpont JR, Mustaphi CJC, Kokelj SV, Smol JP. Arctic climate warming and sea ice declines lead to increased storm surge activity. Geophys Res Lett. 2013; 40(7): 1386–1390. 

77. Öhman MC, Sigray P, Westerberg H. Offshore windmills and the effects of electromagnetic fields on fish. Ambio. 2007; 36(8): 630–633. 

78. Gill AB, Huang Y, Spencer J, Gloyne-Philips I. Electromagnetic fields emitted by high voltage alternating current offshore wind power cables and interactions with marine organisms. Electromagnetics in Current and Emerging Energy Power Systems Seminar. London, UK. COWRIE. 2012.

81. Stenberg C, Støttrup JG, van Deurs M, Berg CW, Dinesen Ge, Mosegaard H, et al. Long-term effects of an offshore wind farm in the North Sea on fish communities. Mar Ecol Prog Ser. 2015; 528: 257–65. 

Tables

Table 3: I suggest reformatting so that the respective data for each storm is easier to follow (i.e. the “Name” column looks like a string of words in a column, it took me a bit to figure out which events related to which values in the rest of the table). 

Table 3: Additional lines added to clarify data across storms and rows.

Figures

General: There are many figures that do not have the “�” before C for temperature. Please incorporate this where it is missing. 

 - “�” before C added to Figures 2, S1.3.a, S1.3.b, and S1.3.c.

Figure 1: Why is the buoy indicated in the map but the data from it never used? Or if the data from it was used, then this was not clear in the manuscript. Also, there should be an inset map showing this region across the broader US East Coast for readers unfamiliar with this region. 

 - Buoy demarcation removed, and subset map included.

Figure 3: There should be a note about the different x-axes. I suggest instead replotting 2016 to be on the time scale as 2017 and 2018 and just begin the data for where its available. 

 - Replotted for consistent x-axis.

Figure 4: Similar comment to above, the x-axes should all be the same. Also the difference in the y-axes should be mentioned in the figure caption. Also the figure caption only includes a description about temperature but needs to also include it for velocity and TKE. 

 - Replotted for consistent x-axis; differing y-axes indicated in caption.

Figure 5: I suggest placing a box around the panel for each event that indicates when the peak winds for the storm occur. This will make it easier for the reader to view the results (which look really nice!) without flipping back and forth between the table with the dates of the storms. 

 - Boxed included.

Figure 6: If the authors are including three events from 2017, why not include all four? It is a bit confusing why one is not included.

 - Rationale for only included first 3 storms included in caption.

Figure 8: I suggest replotting to have all graphs have the same y-axis so they can be compared across the years. Also, the date ranges are confusing for the 2017 panel. It is unclear which dates (and associated box plots) indicate before or after a storm. Also, some of the years have a lot of variance. Is there a reason for this? 

 - Replotted for consistent y-axis range. Figure caption altered to explicitly note grouping of boxes. 

Figure 9: The red circles and the mean behind red dashed and solid lines are not included in the figure caption. Please include this. Also, where is the data for the Northern Site 2018? I realize the model did not converge but it did not for other regions and years yet they have all the data plotted. 

 - Red lines corrected for consistency, and description of circles included. Northern 2018 data included (observed time series); absence of model estimates caused by inability of model to successfully converge (described in Results).

SI1-2: Same comments as for Figure 4

 - Replotted for consistent x-axis.

SI3a-c: It appears there is some model bias with FVCOM on average underestimating the observed temperatures yet this is never addressed in the manuscript. How was this dealt with? In some cases the bias was almost on the order of 5�C which is very significant for fish (e.g. Southern Site 2017). 

 - Bias noted and compared; further elaboration of bias beyond scope of study; note that observed temperature was used for all modeling (not FVCOM-derived temperature).

---

## [Decision Letter · Decision Letter 1]

5 Aug 2020

PONE-D-20-11553R1

The recurring impact of storm disturbance on black sea bass (Centropristis striata) movement behaviors in the Mid-Atlantic Bight.

PLOS ONE

Dear Dr. Wiernicki,

Thank you for submitting your manuscript to PLOS ONE. After careful consideration, we feel that it has merit but does not fully meet PLOS ONE’s publication criteria as it currently stands. Therefore, we invite you to submit a revised version of the manuscript that addresses the points raised during the review process. As well as the recommendations on the content, it is important to address all the minor edits recommended by the reviewer and any additional ones the authors find when going over the manuscript, as PLOS ONE has no in-house editing services. 

We look forward to receiving your revised manuscript.

Kind regards,

Vanesa Magar, Ph.D.

Academic Editor

PLOS ONE

Reviewers' comments:

Reviewer's Responses to Questions

**Comments to the Author**

1. If the authors have adequately addressed your comments raised in a previous round of review and you feel that this manuscript is now acceptable for publication, you may indicate that here to bypass the “Comments to the Author” section, enter your conflict of interest statement in the “Confidential to Editor” section, and submit your "Accept" recommendation.

Reviewer #2: (No Response)

2. Is the manuscript technically sound, and do the data support the conclusions?

Reviewer #2: Yes

3. Has the statistical analysis been performed appropriately and rigorously? 

Reviewer #2: Yes

4. Have the authors made all data underlying the findings in their manuscript fully available?

Reviewer #2: Yes

5. Is the manuscript presented in an intelligible fashion and written in standard English?

Reviewer #2: Yes

6. Review Comments to the Author

Reviewer #2: General Comments/summary:

Major comments: It is clear that the authors put time, effort, and thought into their manuscript edits and this revised manuscript is much improved when compared to the first version. Especially in the introduction, methods and results, the messages are much clearer and easier to follow than before. One issue I continue to have is that the authors have chosen to still discuss wind energy in the context of storm disturbance but do not full develop the argument as to why and how we should be concerned about wind energy and storms. The major issue is that in the abstract and in the discussion the authors are vague as to whether the impacts of recurring storms will act synergistically with impacts of offshore wind development or if the information gained from studies like this will provide information as to how black sea bass will respond to the disturbance from offshore wind. I tried to provide edits throughout that may guide the authors as the mentions of wind energy still read as a tangential issue to the concerns in this study. The discussion still needs some considerable editing (see in specific comments). Finally, especially in the newly revised sections, there are still areas that need general editing, mostly for grammar. I tried to point this out where it occurred but I suggest all authors re-read and edit the manuscript before resubmission.

7. PLOS authors have the option to publish the peer review history of their article (what does this mean?). If published, this will include your full peer review and any attached files.

Reviewer #2: No

---

## [Author Response · Author response to Decision Letter 1]

14 Sep 2020

Dear Dr. Magar:

Thank you for your very thorough second review. We greatly appreciate your commitment to improving this manuscript. Again, we have addressed all general and specific comments below, referencing line numbers for more substantive changes in the clean/marked copy.

Sincerely Yours,

Caroline Wiernicki, NOAA Knauss Fellow 

cc: M O’Brien

 F Zhang

 V Lyubchich

 M Li

 D Secor

Review of Wiernicki et al. THE RECURRING IMPACT OF STORM DISTURBANCE ON BLACK SEA BASS (CENTROPRISTIS STRIATA) MOVEMENT BEHAVIORS IN THE MID-ATLANTIC BIGHT. Submitted to PLoS ONE. Research Article 

General Comments/summary:

Major comments: It is clear that the authors put time, effort, and thought into their manuscript edits and this revised manuscript is much improved when compared to the first version. Especially in the introduction, methods and results, the messages are much clearer and easier to follow than before. One issue I continue to have is that the authors have chosen to still discuss wind energy in the context of storm disturbance but do not full develop the argument as to why and how we should be concerned about wind energy and storms. The major issue is that in the abstract and in the discussion the authors are vague as to whether the impacts of recurring storms will act synergistically with impacts of offshore wind development or if the information gained from studies like this will provide information as to how black sea bass will respond to the disturbance from offshore wind. I tried to provide edits throughout that may guide the authors as the mentions of wind energy still read as a tangential issue to the concerns in this study. The discussion still needs some considerable editing (see in specific comments). Finally, especially in the newly revised sections, there are still areas that need general editing, mostly for grammar. I tried to point this out where it occurred but I suggest all authors re-read and edit the manuscript before resubmission. 

Thank you for your careful review. Regarding major comments on the argument on wind energy concerns: we have followed your guidance and incorporated more specific dialogue linking uncertainties from wind farm construction disturbance relative to storm disturbance, and how the interaction may impact local abundance for Mid-Atlantic fisheries. Regarding concerns over general edits to grammar and flow, all authors have re-read the manuscript and provided edits.

Thank you once again for the attention to our manuscript. We have endeavored to match this in our revisions. For brevity, our responses to specific comments are fairly direct.

Author response to specific comments/edits on text, tables, and figures:

Abstract 

Line 24: I suggest adding some statement after Middle Atlantic Bight such as “along the U.S. Northeast” or “in the Northwest Atlantic” so that readers outside of the U.S will know where this study takes place. Finally, remove the abbreviation because the MAB abbreviation is used only one other time (Line 28).

 - Lines 24-25: Phase “in the Northwest Atlantic” added to clarify broader geographic area of interest.

 - Lines 28-29: Abbreviation “MAB” replaced with “Middle Atlantic Bight”.

Line 40: “…, effects.”

 - Line 41: “effect” pluralized. 

Line 42: “US”. The authors have not defined the abbreviation of “US”, which may be obvious to some, but should be avoided for readers outside of the United States.

 - Line 43: “US” replaced with “United States”.

Lines 42-44: This sentence is vague and not well written, so I am unsure what message it is trying to convey. What do the authors mean “in context of storm as a recurring natural disturbance”? This is where a distinction between understanding natural disturbances to aid in predictions of future anthropogenic disturbances vs. concerns with confounding effects of storms and wind energy should be made. 

 - Lines 43-45: Sentence rephrased, “Their availability to fisheries surveys and sensitivity to wind turbine impacts will be biased during periods of high storm activity, which is likely to increase with regional climate change.”

Line 64: “…tighter coupling of fish to structured habitat.”

 - Line 73: Term “habitat” added.

Line 64-65: Remove “Thus, the literature supports that severe storms can be disruptive events to fish communities” and insert the sentence beginning with, “Thus, the vast majority of storm impacts on marine communities…” and remove “Still,” and replace with “and” to combine these sentences.

 - Lines 73-75: Sentence rephrased: “Marine community responses to even single storm events represents challenging field science, perhaps contributing to a lack of studies on multiple storm events as a recurring source of natural disturbance.”

Lines 105-106: Remove “, including black sea bass” as black sea bass as a focal study species has not been introduced yet. 

 - Line 121: Removed “, including black sea bass”.

Line 114: Perhaps at the end of this sentence the authors can include some information about offshore wind development and the impacts it can have on fish. Here the authors can introduce for the first time whether they are discussing offshore wind as an additive effect to storms or that storm disturbance studies can be used to understand the impacts of offshore wind development. 

 - Line 146-177-131: Sentences added, “Comprehensive understanding of this natural disturbance regime shaped by storms is both timely and relevant to the MAB, as the region is currently undergoing evaluation by industry and policy stakeholders for offshore wind energy development [43]. Black sea bass is a model species for understanding wind energy impacts because they are ubiquitous and commercially important within the > 7 103 km2 of leased US Federal waters (https://www.boem.gov/renewable-energy/state-activities).. To best utilize this and similar demersal species in understanding both negative and positive impacts of offshore wind energy development, baseline information is needed on storm impacts to black sea bass. Storm effects on fish behavior are a pervasive recurrent natural disturbance in the MAB, which if not understood and accounted for, will likely bias wind turbine impact studies.”

Line 122-123: “…characteristically occupy shelf natural …to late October”. This phrase is redundant from the one above in Line 120. Consider editing. 

 - Line 138: Sentence rephrased, “…nearshore shelf habitats…”

Materials and Methods

Line 179-180: The information about how to sex black sea bass needs to be cited. Also it is unclear if the authors are visually inspecting after they make an incision for the tag, if they are sexing via a cannula, or if they are merely using abdominal pressure to expel milt or eggs (especially as they seemed to have issues with sexing a good amount of fish and I am curious as to why). Finally, the authors should make a reference that their collection period overlaps with around the peak spawning time. 

 Line 230-232: Additional text added to clarify procedure for determining sex, “Sex was determined by the visual inspection and identification of gonads, which were visible during surgery through the incision (see below) [44].”

 - Line 232: Citation added for sex determination methodology.

Lines 262: Remove “those” and add “data” before “points”

 - Line 325: Removed “those” and added term “data”.

Lines 270-271: “This analysis was performed in R, …”

 - Line 340: Removed “carried out” and replaced with “performed”.

Lines 276-277: “The predictors were tested for their influence on the response variable, daily average movement, and included daily average TKE, …”

 - Lines 345-346: Terms added for clarification: “The predictors were tested for their influence on the response variable, daily average movement index, and included daily average TKE, observed…”

Discussion

Line 509 and 520: These paragraphs are slightly redundant; consider combining. 

 - Lines 589-595: Paragraphs combined; sentence removed as it was redundant, “Results also indicated that multiple storm events in a given year may precondition the impacts of subsequent storms on water column stratification, though we failed to detect a cumulative impact of repeated storms on black sea bass movement.” 

Line 535-539: “Biotelemetry detections … permanent evacuations.” Either remove this information or add it to the limitations paragraph. 

 - Line 620: Sentence removed, “Biotelemetry detections can be biased low during storm events when ambient noise interferes with detection of transmitted signals; we conducted analyses robust to this source of bias through an ARIMA intervention analysis, and observed that, in all years, late-season storms were associated with permanent evacuations.”

Line 539: Remove “where” and replace with “when”

 - Line 620: Replaced “where” with when”.

Line 545-547: “This suggestions … in movement behavior.” This is an important result and is buried in text within the paragraph. I suggest moving this information to the beginning of the paragraph to highlight the authors important findings. 

 - Lines 618-620: Information moved earlier in paragraph, “Furthermore, analysis of storm-driven environmental variables during the multiple-storm year, 2017, indicated storm-driven destratification was the primary catalyst for evacuations.” 

Line 551: Remove “of” and replace with “to”. 

 - Line 645: Replaced “of” with “to”.

Line 564 and Line 566: Remove “magnitude in” and make temperature plural 

 - Lines 658-659: Removed “magnitude in” and made “temperature” plural.

Line 568: “Our findings …”

 - Line 662: “Our” added.

Line 577: This paragraph needs editing. The impacts on population dynamics and what the authors mean by population dynamics is vague. Perhaps the other paragraph later in the discussion about black sea bass reproduction can be incorporated into this paragraph. 

 - Line 675-686: Sentences rephrased/added, “The two behavioral responses measured in this study—evacuations or decreased movement in the face of acute disturbance to habitat conditions—are distinct modes, which may have carryover effects to feeding, reproduction, and predator evasion. Black sea bass occupy small home ranges on structured habitats (0.14-7.4 km2) [40], where they feed on reef and adjacent seabed prey items [38, 63, 64]. Reproduction has been observed to occur primarily during June- September [38] and occurs frequently with spawning intervals for females estimated to be 2.7 to 4.6 days [65] . Thus late-summer and early-fall storms, which caused depressed movements away from structure likely interfere with feeding and courtship, which occurs in regions adjacent to the reef. The alternate behavior, evacuation, likely disrupts mating systems and feeding territories and incurs greater predation risk. Additional research is called for on changed feeding and reproductive states before and after storms, the fate of evacuees, and whether evacuations contribute to the greater fall seasonal migration to deeper shelf habitats.”

Line 578: “..are distinct”. Distinct responses? Behaviors? 

 - Line 676: Term “modes” added.

Lines 582-583: Edit to “late-summer fall, may impact habitat use…”

 - Lines 675-686: Original phrase removed.

Line 587: Change “don’t” to “do not”.

 - Line 675-686: Original phrase removed

Lines 588-589: This topic sentence needs editing. What do the authors mean by inferences? 

 - Line 688: Phrase included for clarification, “…on movement behaviors…”

Line 597: Edit to “…relate to study design include assumptions…” 

 - Line 761: Sentence edited, “Key limitations to our findings related to study design include…”

Line 615: This paragraph either needs to be developed more or incorporated with another paragraph (see above). 

 - Lines 675-686: Paragraph edited and features incorporated into earlier discussion on differential impacts to population dynamics.

Line 619: The Drohan et al 2007 citation is wrong; the studies that show late season spawning are included within the Drohan paper. 

 -Line 680: Drohan citation switched with Berrien and Sibunka 1999.

Line 620: The potential impacts on black sea bass spawning are a stretch and need to be clarified that this late season spawning is rare and the typical spawning season is earlier in the summer and would not necessarily be affected (but maybe during years like 2017). The papers within the Drohan paper that claim late season spawning are old papers and estimate black sea bass spawning through egg and larval black sea bass surveys. 

 - Line 679-681: Additional information on spawning included, particularly that black sea bass are frequent spawners so events that cause several d periods of depressed activity likely impact courtship and reproduction, “Reproduction has been observed to occur primarily during June-September [38] and occurs frequently with spawning intervals for females estimated to be 2.7 to 4.6 days [65] .” 

Line 627: It is unclear why storms would lead to latitudinal range shifts. Either develop this concept more or remove it. 

 - Line 793: “latitudinal range shifts” removed.

Line 631: “erected” is an awkward word choice.

 - Line 797: Replaced “erected” with “established”. 

Line 632: I suggest reordering fishing and wind farm impacts to reflect the order in which these are discussed in the paragraph 

 - Line 798: Order of wind farm and fishing impacts switched to reflect discussion in paragraph.

Line 631-636: Here is where the authors need to clarify because section is vague as to whether lessons from this study can aid in predictions of construction impacts or if we need to think about storms and construction impacts together. If I understand correctly, wind farm development and increasing storm impacts could lead to lower abundances of fish in those regions and this needs to be considered in the context of fisheries. 

 - Lines 865-866: Sentence added, “Furthermore, collective impacts from wind farm construction coupled with impacts from natural storm disturbance could lead to altered fish abundances in regions along the cold pool front.”

Line 633: Edit to “…will occur during periods and in regions…”

 - Line 799: Terms “during” and “in” added accordingly.

Line 637-639: “Each of these stresses can…construction alone.” How? Are there examples? Is this speculation? Obscuring and enhancing impacts are opposite effects, so is there a potential storms interacting with wind farms is beneficial? 

 - Lines 805-861: Additional sentence added to paragraph for clarification, ““Refuge- seeking behaviors associated with pile-driving or vessel noise may be similar in kind to depressed movements associated with storms. Storm-induced evacuations could obscure departures associated with wind turbine construction. Similarly, reduced catchability could be erroneously associated with wind turbine impacts following a period of high storm activity.”

Line 640: “…catchability in particular.” The authors after here transition to anecdotal information about how storms can reduce catchability yet the above sentence is in reference to pile driving. So now it would appear the authors are using the impacts of storms as a way to understand the impacts of offshore wind development, where before they were discussing the effects acting in concert with one another. 

 - Lines 862-872: Thanks, we have altered text so reduced catchability is not linked to pile driving but rather use the last paragraph to speculate how wind farm and storm impacts could jointly impact distributional changes. “The nature of the potential interaction of natural storm disturbance with wind farm construction disturbance is not well known, and future research on whether this interaction is beneficial or detrimental to fish abundance in affected regions is critical for future management. Furthermore, collective impacts from wind farm construction coupled with impacts from natural storm disturbance could lead to altered fish abundances in regions along the cold pool front. Anecdotal reports from charter fishers suggest greatly reduced catch rates following major storms (D. Zemeckis, Rutgers University, pers. comm.). Storms may also catalyze fall departures to deeper shelf environments shifting fisheries and influencing their accessibility to bottom trawl surveys [85]. The need for storm disturbance to be incorporated in baseline and impact monitoring is heightened by the prediction of higher occurrence of higher intensity storms related to climate change.”

---

## [Editor Report · Decision Letter 2]

17 Sep 2020

The recurring impact of storm disturbance on black sea bass (Centropristis striata) movement behaviors in the Mid-Atlantic Bight.

PONE-D-20-11553R2

Dear Dr. Wiernicki,

We’re pleased to inform you that your manuscript has been judged scientifically suitable for publication and will be formally accepted for publication once it meets all outstanding technical requirements.

Kind regards,

Vanesa Magar, Ph.D.

Academic Editor

PLOS ONE
---

## [Editor Report · Acceptance letter]

25 Sep 2020

PONE-D-20-11553R2 

The recurring impact of storm disturbance on black sea bass (*Centropristis striata*) movement behaviors in the Mid-Atlantic Bight 

Dear Dr. Wiernicki:

I'm pleased to inform you that your manuscript has been deemed suitable for publication in PLOS ONE. Congratulations! Your manuscript is now with our production department. 

Kind regards, 

on behalf of

Dr. Vanesa Magar 

Academic Editor

PLOS ONE